



# Evaluating daytime planetary boundary-layer height estimations resolved by both active and passive remote sensing instruments during the CHEESEHEAD19 field campaign

James B. Duncan Jr.[1,2*], Laura Bianco[1,2], Bianca Adler[1,2], Tyler Bell[3], Irina V. Djalalova[1,2], Laura Riihimaki[1,4], Joseph Sedlar[1,4], Elizabeth N. Smith[5], David D. Turner[6], Timothy J. Wagner[7], and James M. Wilczak[2]

[1] University of Colorado/Cooperative Institute for Research in Environmental Sciences, Boulder, 80305, CO, USA
[2] National Oceanic and Atmospheric Administration, Physical Science Laboratory, Boulder, 80305, CO, USA
[3] University of Oklahoma, Norman, 73071, OK, USA
[4] National Oceanic and Atmospheric Administration, Global Monitoring Laboratory, 80305, Boulder, CO, USA
[5] National Oceanic and Atmospheric Administration, National Severe Storms Laboratory, Norman, 73072, OK, USA
[6] National Oceanic and Atmospheric Administration, Global Systems Laboratory, Boulder, 80305, CO, USA
[7] Space Science and Engineering Center, University of Wisconsin–Madison, Madison, 53806, WI, USA
[*] Now at WindESCo, Burlington, 01803, MA, USA

*Correspondence to*: Laura Bianco (Laura.Bianco@noaa.gov)

**Abstract.** During the Chequamegon Heterogeneous Ecosystem Energy-balance Study Enabled by a High-density Extensive
Array of Detectors 2019 (CHEESEHEAD19) field campaign, held in the summer of 2019 in northern Wisconsin, U.S.A., active and passive ground-based remote sensing instruments were deployed to understand the response of the planetary boundary layer to heterogeneous land surface forcing. These instruments include Radar Wind Profilers, Microwave Radiometers, Atmospheric Emitted Radiance Interferometers, Ceilometers, High Spectral Resolution Lidars, Doppler Lidars, and Collaborative Lower Atmospheric Modelling Profiling Systems that combine several of these instruments. In this study,
these ground-based remote sensing instruments are used to estimate the height of the daytime planetary boundary layer, and their performance is compared against independent boundary-layer depth estimates obtained from radiosondes launched as part of the field campaign. The impact of clouds (in particular boundary layer clouds) on boundary-layer depth is also investigated.

We found that while overall all instruments are able to provide reasonable boundary-layer depth estimates, each of them shows
strengths and weaknesses under certain conditions. For example, Radar Wind Profilers perform well during cloud free conditions, and Microwave Radiometers and Atmospheric Emitted Radiance Interferometers have a very good agreement during all conditions, but are limited by the smoothness of the retrieved thermodynamic profiles. The estimates from Ceilometers and High Spectral Resolution Lidars can be hindered by the presence of elevated aerosol layers or clouds, and the



multi-instrument retrieval from the Collaborative Lower Atmospheric Modelling Profiling Systems can be constricted to a

limited height range in low aerosol conditions.

## 1 Introduction

The Chequamegon Heterogeneous Ecosystem Energy-balance Study Enabled by a High-density Extensive Array of Detectors 2019 (CHEESEHEAD19) field campaign, held between the mid-summer and fall of 2019, investigated the surface energy balance and atmospheric response over the heterogeneous forest region of northern Wisconsin, U.S.A. (Butterworth et al.,

2021). An extensive array of instrumentation was deployed by the National Science Foundation (NSF), the National Oceanic and Atmospheric Administration (NOAA), other agencies, and universities to examine the impacts of land-surface heterogeneities within the forested region on planetary boundary-layer (PBL) structure and evolution. An essential component of CHEESEHEAD19 was to adequately sample and model via large eddy simulation the spatial variability within the study region, so a correct understanding of subgrid-scale processes could be used to improve the performance of numerical weather

and climate prediction models. The details and the design of the field campaign including the complete list of instruments deployed and a discussion of preliminary results are presented in Butterworth et al. (2021). The complete dataset of observations and model runs is available for general use through the National Center for Atmospheric Research (NCAR) Earth Observatory Laboratory (EOL) data repository ([www.eol.ucar.edu/field_projects/cheesehead](www.eol.ucar.edu/field_projects/cheesehead)).

Additional studies using the CHEESEHEAD19 dataset are in progress. These studies include an observational investigation

into the role of boundary layer clouds on the partitioning of the surface energy budget during daytime and PBL structure, evaluation of numerical weather prediction models and satellite representations of clouds and surface radiation, and an observational analysis of the forcing mechanisms in the PBL evolution and structure in this particular geographical area.

The planetary boundary layer height (PBLH) plays an important role in determining local air quality, turbulent transport of

heat and moisture, and is a key parameter in model parameterizations. Land surface heterogeneities contribute to variability in PBL structure and depth (Desai et al., 2006; Reen et al., 2014; Gantner et al., 2017; Platis et al., 2017). Many studies (some of which listed later in the manuscript) have made use of different observational datasets to derive the PBLH. In this study, an intercomparison analysis of the strengths and weaknesses of the different platforms deployed for CHEESEHEAD19 at estimating daytime PBLHs compared to radiosonde estimates is presented. The instruments deployed during

CHEESEHEAD19 and used to this purpose include the Atmospheric Emitted Radiance Interferometer (AERI), the Microwave Radiometer (MWR), the Vaisala CL51 Ceilometer (CL51), the High Spectral Resolution Lidar (HSRL), the Collaborative Lower Atmospheric Modelling Profiling System (CLAMPS multi-instrument), and the 915-MHz Radar Wind Profiler (RWP). Thermodynamic profiles in the PBL can be retrieved from passive multi-spectral radiance observations from instruments such as the AERI and the MWR (Turner and Blumberg, 2019), and therefore, instruments such as these can be used to estimate

PBLH using standard methodologies (i.e., parcel method, maximum vertical gradient of the potential temperature, etc.). Cimini



et al. (2013) previously made use of direct observations of MWR brightness temperatures instead of retrieved profiles to estimate PBLH. However, this approach, which investigates the covariance of geophysical variables, requires an independent PBLH reference observation for training, which was not available during CHEESEHEAD19.

Ceilometer backscatter is correlated with aerosol concentrations at a given height. The transition to the free atmosphere, and therefore the PBLH, can be inferred as the height above which backscatter intensity strongly decreases with height (Hicks et al., 2016). This study evaluates ceilometer PBLH estimates based on proprietary software which leverages this information to provide PBLH estimates (e.g., Münkel et al., 2007). Alternative algorithms exist to extract PBLH estimates from ceilometer data, such as using covariance wavelet transforms to identify the location of the peak negative gradient (e.g. Brooks, 2003; Morille et al., 2007; Compton et al., 2013). Some of these methods have been evaluated against the standard PBLHs provided by the proprietary software (Knepp et al., 2017) finding improvements at some sites and in some particular situations (i.e. morning transition), but not on a consistent basis. Future research will be specifically devoted to identifying and testing more refined techniques to estimate PBLH from ceilometer observations.

The HSRL is an advanced lidar system that measures both the elastic backscatter (similar to a ceilometer) and the inelastic backscatter that is entirely due to molecular (and not particulate) scattering (e.g., Eloranta, 2005). From these two observations, an aerosol backscatter coefficient unattenuated by molecular scattering can be derived. Furthermore, HSRLs are often designed to have very good solar background suppression, yielding much larger signal-to-noise ratio data from which higher order moments can be derived (McNicholas and Turner, 2014). The often-present sharp gradient in aerosol backscatter at the top of the PBL during the daytime can be used to determine the PBLH.

Doppler lidars (instrumented within the CLAMPS) can also be used to estimate PBLHs. Bonin et al. (2018) used a fuzzy logic-based approach to combine the information derived from Doppler lidar-measured wind and turbulence profiles (vertical velocity and its variance, signal-to-noise ratio and its variance) for PBLH characterization. Since Doppler lidar observations may not always reach the top of the PBL, thereby limiting their ability to detect the PBLH (Tucker et al., 2009; Berg et al., 2017, Bonin et al., 2018), Krishnamurthy et al. (2021) utilized a machine learning framework for estimating PBLH using parameters derived from Doppler lidars and surface meteorological measurements that accounts for this limitation.

RWPs are also used to measure PBLH as the signal-to-noise ratio (SNR) exhibits a local maximum at the height of the inversion, due to small-scale buoyancy fluctuations associated with the entrainment process (White, 1993; Angevine et al., 1994; Coulter and Holdridge, 1998). Other information contained in the hourly variance and spectral width of the RWP-measured vertical velocity can be included to strengthen the robustness of PBLH determination (Bianco et al., 2008). Daytime PBLH values derived from a 915-MHz RWP have additionally been shown to agree well with the maximum in the variance of the water vapor mixing ratio, which provides another measure of PBLH (Turner et al., 2014).

Radiosondes are often considered the gold standard for retrieving PBLH due to their high vertical resolution and the different meteorological variables that can be measured. However, a limitation of radiosondes is that they normally do not provide the high-temporal resolution that is possible from remote sensing systems, such as those deployed for CHEESEHEAD19. If remote sensing systems are to be used to provide high temporal resolution estimates of the PBLH, a comprehensive understanding of



how these ground-based remote-sensing instruments resolve the PBLH is needed in order to accurately interpret the retrieved PBLHs.

In this study, 170 daytime radiosonde observations collected during the field campaign are used to validate the performance of the aforementioned instruments deployed for CHEESEHEAD19 in retrieving PBLHs. Since not all instruments were deployed for the entire duration of the campaign (see Section 2 for details), evaluation of these instruments in their ability to

resolve PBLH will be broken down into two components. First, the RWP, CL51, and MWR PBLH estimates will be compared with the radiosondes over a multi-month time period. Second, the analysis will focus on two of the short-term intensive observation periods (IOPs), when a larger number of instruments are available. The analysis also includes PBLH comparisons in different types of cloud conditions using RadSys measurements of incoming and outgoing radiation and cloud properties. Section 2 provides the CHEESEHEAD19 deployment specifics, including instrument description and the instrument-specific

methods used to derive PBLHs; in Section 3 the estimation and characterization of the PBLHs from RWPs, CL51s, and MWRs is evaluated against radiosonde estimates for the multi-month time period analysis, from all available platforms against radiosonde estimates for the two IOPs, and in more detail from two single-day case studies. Section 4 concludes with a summary of the results and provides concluding remarks.

## 2 CHEESEHEAD19 Field Campaign Instrumentation and Deployment Specifics

Many of the instruments deployed for CHEESEHEAD19 were located within the main 10 km×10 km experiment domain centred on the existing Park Falls 447-m-tower AmeriFlux/NOAA supersite, WLEF (see Fig. 1 of Butterworth et al., 2021, for the map and schematic diagram of the CHEESEHEAD19 domain). In addition to instrumentation deployed at the WLEF site, some instruments were deployed at the Prentice and Lakeland airports, located approximately 45 km south and east of the WLEF site, respectively (Fig. 1a), to provide information on the spatial variability of boundary-layer structure and cloud and

radiation fields.





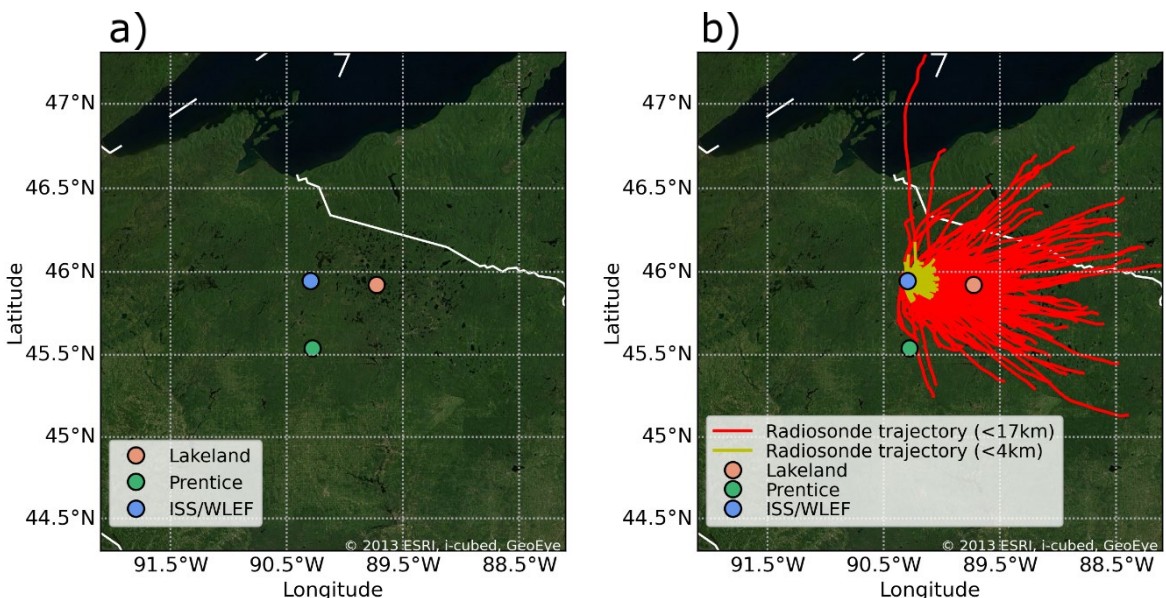

**Figure 1: a) Location of the four measurement sites where instruments were deployed to examine PBL structure and evolution during the CHEESEHEAD19 field campaign. The background shows aerial imagery. Since the ISS and WLEF sites were located less than 2 km apart, they are plotted with the same marker. b) Trajectories of all the radiosondes released from the ISS facility (red up to 17 km agl, yellow up to 4 km agl) are shown.**

One AERI and the HSRL (data available from Wagner, 2020) were deployed aboard the Space Science and Engineering Center (SSEC) Portable Atmospheric Research Center (SPARC, Wagner et al., 2019); two MWRs (data available from Adler et al., 2021), two CLAMPS platforms (Wagner et al., 2019; data available from Klein et al., 2020), two CL51s (data available from Sedlar et al., 2020a, b), and two RWPs (data available from Bianco and Duncan, 2020, and Wilczak and Gottas, 2020) were deployed at the Lakeland and Prentice sites. The radiosondes were launched from the Integrated Sounding System (ISS) site, located slightly west (< 2 km) of the WLEF site. Throughout the campaign, routine soundings were launched at 13:00 local time (LT; LT = UTC - 5). In addition to these launches, there were three 7-day IOPs wherein three to four additional launches were made per day, allowing for a more complete picture of the PBLH development.

The list of the instruments used in this study, their location, and their deployment duration is presented in Table 1.

| Instrument System | Deployment Locations | Deployment Period (mm/dd) | Used in the Multi-Month Time Period Analysis | Used in the Summer-IOP Analysis | Used in the Fall-IOP Analysis |
|---|---|---|---|---|---|
| Radiosondes | ISS | 6/20–10/11 | X | X | X |
| AERI | Lakeland | 9/20–10/10 | | | X |





| | | | | | |
|---|---|---|---|---|---|
| | WLEF | 7/2–9/11 | | X | |
| Radiometrics MP-3000A MWR | Lakeland | 8/1–10/29 | X | X | |
| | Prentice | 7/31–9/18 | X | | |
| RPG HATPRO-G4 MWR | Lakeland | 9/20–10/22 | X | | X |
| CLAMPS multi-instrument | Lakeland | 9/20–10/1 | | | X |
| | Prentice | 9/20–10/9 | | | X |
| CL51 | Lakeland | 6/27–10/22 | X | X | X |
| | Prentice | 6/28–10/19 | X | | |
| RWP | Lakeland | 6/27–10/29 | X | X | X |
| | Prentice | 6/26–10/30 | X | | |
| HSRL | WLEF | 6/24–10/06 | | X | X |

**Table 1: List of instruments used in this study, with site and time of the deployment during CHEESEHEAD19, and information indicating the analysis time periods.**

The methods used by these instruments to discern PBLH development are described in the subsequent subsections and vary from simple methods, such as the parcel method, to more sophisticated instrument-specific techniques.

### 2.1 Validation Dataset - Radiosondes

A total of 170 Vaisala RS41 radiosondes were launched during daytime from the ISS location between 20 June and 11 October 2019. The soundings were typically launched at least once per day at 13:00 LT. In addition to these daily soundings, three 7-
day IOPs occurred between 9-13 July, 19-24 August, and 23-28 September. Due to the larger instrument availability, only the latter two IOPs are used in this study. During these two IOPs, four radiosondes were launched per day at around 06:00, 09:00, 13:00, and 17:00 LT. These IOPs allow for the diurnal development of the PBL to be resolved (early morning, morning transition, midday, and evening transition), while the more standard launch time of 13:00 LT enabled the examination of the midday, and presumably more maturely-developed (although not necessarily well-mixed) PBL.



The radiosondes were quality controlled and are maintained by the NCAR EOL. The Vaisala system performs a sequence of standard quality control procedures and corrections for all radiosonde data. In addition to the standard Vaisala procedures, custom quality control was used to mitigate any previously unidentified issues. Specific quality control measures imposed are detailed in the NCAR EOL technical report (NCAR/EOL, 2019).

Radiosonde trajectories for these launches up to 17 km above ground level (agl) are displayed in red in Fig. 1b. Generally, the
radiosondes track to the east. The yellow lines in Fig. 1b shows the trajectories up to 4 km agl and demonstrate that each radiosonde launch was within the investigation area and are appropriate for the PBLH instrument evaluation analysis performed later.

Radiosonde profiling capabilities allow for a variety of methods to be used to estimate the PBLH (Seibert et al., 2000; Seidel et al., 2010), each of which present different strengths in resolving the PBLH (Seidel et al., 2010; Li et al., 2021). These
include:

1. Parcel method: The PBLH is evaluated by comparing the surface value of virtual potential temperature ($\Theta_{v, surf}$) to values aloft and identifying the height where $\Theta_v$ is the same as the surface value. This is the height where a parcel of air, lifted from the surface, is in equilibrium with its environment at this altitude (Holzworth, 1964). In this study, a slightly different criterion (i.e, modified parcel method) is used where the PBLH is defined as the height at which $\Theta_v$ is the same as the $\Theta_{v,}$
$_{surf}$ value at the surface $+0.5 K$ ($\Theta_v = \Theta_{v, surf} + 0.5 K$). This definition is similar to that used in some operational numerical weather prediction models (Coniglio et al., 2003), whose verification and validation is the goal of a CHEESEHEAD19 study in progress.

2. Potential temperature gradient method: The PBLH is evaluated by finding the location of the maximum vertical gradient of potential temperature ($\Theta$) in the lowest 4 km agl.

3. Specific humidity gradient method: The PBLH is evaluated by locating the height of the minimum vertical gradient of specific humidity ($q$) in the lowest 4 km agl.

4. Relative humidity gradient method: The PBLH is evaluated by locating the height of the minimum vertical gradient of relative humidity ($RH$) in the lowest 4 km agl.

5. Elevated inversion method: The PBLH is evaluated by locating the height of the base of an elevated temperature ($T$)
inversion.

Using the methods listed above, Fig. 2 provides a comparison between the distribution of the PBLH estimation technique over different hours of the diurnal cycle from the radiosondes.





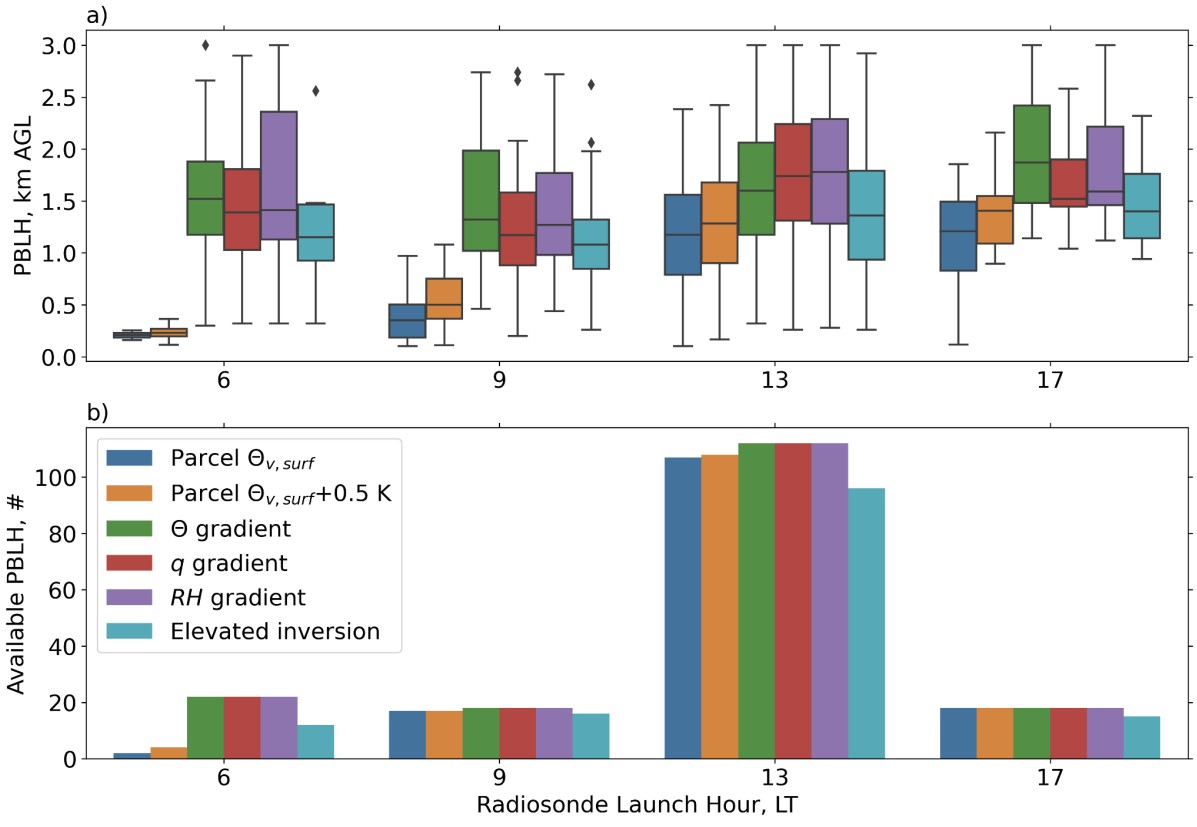

**Figure 2: a) Method-specific PBLH estimates derived from the radiosondes launched during CHEESEHEAD19 as a function of the radiosonde launch hour. b) Number of available PBHL estimates per radiosonde launch hour per method.**

Comparison between these methods demonstrates the parcel method's unique ability (blue and orange bars in Fig. 2a, using either of the two definitions) to detect the morning development of the PBL. Other methods during the morning transition primarily detect the top of a residual layer aloft. Due to stable stratification remaining from the night time hours, fewer PBLH estimates are provided by the early morning radiosonde launches (06:00 LT, Fig. 2b) when applying the parcel method compared with the other methods. Nevertheless, because 1) Seibert et al. (2000) suggested that parcel methods are better suited

in convective boundary layer conditions, and 2) because the parcel method is the only method to reasonably resolve the PBL development, the modified parcel method (hereafter simply referred to as the "parcel method") is employed to derive PBLH from the radiosonde, AERI, and MWR $\Theta_v$ profiles.





## 2.2 Passive Remote Sensing Instruments

As opposed to active remote sensing instruments, which actively send out a signal and record the measured backscatter, passive remote sensing instruments naturally measure existing atmospheric energy or signal to determine quantities of interest via some retrieval algorithm (see Maahn et al., 2020, for a high-level discussion of retrievals and their uncertainties). There were several passive ground-based systems deployed during the field campaign, including the two AERIs and three MWRs used in this study.

### 2.2.1 AERI and MWR

The AERI is a ground-based passive spectrometer receiving downwelling infrared radiation. The spectral resolution of the AERI is better than one cm-1 between the wavelengths of 3.3 and 18.2 $\mu$m (550-3000 cm-1) (Knuteson et al., 2004). The AERI observes the spectrally resolved downwelling radiance approximately every 30 s; however, these high-temporal resolution data typically have a noise filter applied (Turner et al., 2006). Thermodynamic retrievals are processed at 5- or 10-
min resolution due to the computational expense of the employed retrieval algorithm (Turner and Blumberg, 2019). The data from the AERIs deployed during CHEESEHEAD19 at the Lakeland and WLEF sites are used in this study. The AERI at Lakeland was deployed as part of the CLAMPS during a portion of the field campaign (see more on the CLAMPS platform in the next section). The SPARC AERI was deployed at WLEF for the duration of CHEESEHEAD19, but an instrument failure likely caused by a nearby lightning strike resulted in no data being available from 12 September 2019 to the end of the
experiment. As a result, WLEF AERI observations are only present for the August IOP analysis and not the September or multi-month time period analyses.

At both Lakeland and Prentice, Radiometrics MP-3000A MWRs (Solheim et al., 1998) were deployed. This instrument passively measures brightness temperature approximately every 2.5 min at 21 frequencies along the 22.2 GHz water vapor line and 14 frequencies along the 60 GHz oxygen absorption band. These brightness temperature measurements are made at
the zenith angle (90°) and at two oblique scan angles (19.8° and 160.2°). The data collected by the RPG HATPRO-G4 MWR (Rose et al., 2005) deployed at the Lakeland site in conjunction with the CLAMPS are available for a shorter time period and will also be included in the analysis. The HATPRO MWR measures downwelling microwave radiance at 7 frequencies along the 22.2 GHz water vapor line and 7 frequencies along the 60 GHz oxygen absorption band at 1 s temporal resolution. The measurements are taken at zenith and at eight oblique scan angles around 18°, 24°, 30°, 42°, 138°, 150°, 156°, and 162°.

Despite the technical differences between these instruments, the measurements of each instrument (infrared radiance for the AERI and microwave radiance for the MWR) can be used in the same physical-iterative retrieval algorithm, TROPoe (formerly AERIoe, Turner and Löhnert, 2014; Turner and Blumberg, 2019) to extract thermodynamic profiles. Utilizing a common algorithm for thermodynamic profile retrievals helps simplify analysis of both the results and their uncertainties.

The retrieved thermodynamic profiles (extracted at 10-min resolution for both the AERI-based and MWR-based retrievals)
are averaged together between ±30 min of the hour, to determine an hourly mean thermodynamic profile, from which the





PBLH is determined using the same parcel method technique as on radiosonde thermodynamic profiles (described in Section 2.1). Hourly PBLH estimates are determined between sunrise and sunset. The $\Theta_v$ profiles were reviewed for plausibility before estimating the PBLH; some AERI retrieved profiles were manually eliminated based on visual inspection. Also, a glitch in the data acquisition system resulted in several days of erroneous data from the Radiometrics MWR at Lakeland which were

removed from the analysis.

Figure 3 presents an example of time–height cross-sections of retrieved $\Theta_v$ profiles at the Lakeland site from the AERI (Fig. 3a) and one of the MWRs (Fig. 3b) on 28 Sep 2019. The vertical grid spacing and temporal resolution are the same for both instruments. Overlaid on both panels are the corresponding PBLH estimates, obtained by the application of the parcel method. From Fig. 3a suspicious retrievals can be noted for the AERI before 06:00 LT, when clouds with liquid water path larger 75 g

m$^{-2}$ were observed in the area. To limit these circumstances, a threshold was applied on the indicator for the convergence of the AERI retrievals, which filtered out most of the suspicious profiles, albeit not all of them. The presence of these clouds doesn't seem to impact the MWR retrievals (Fig. 3).

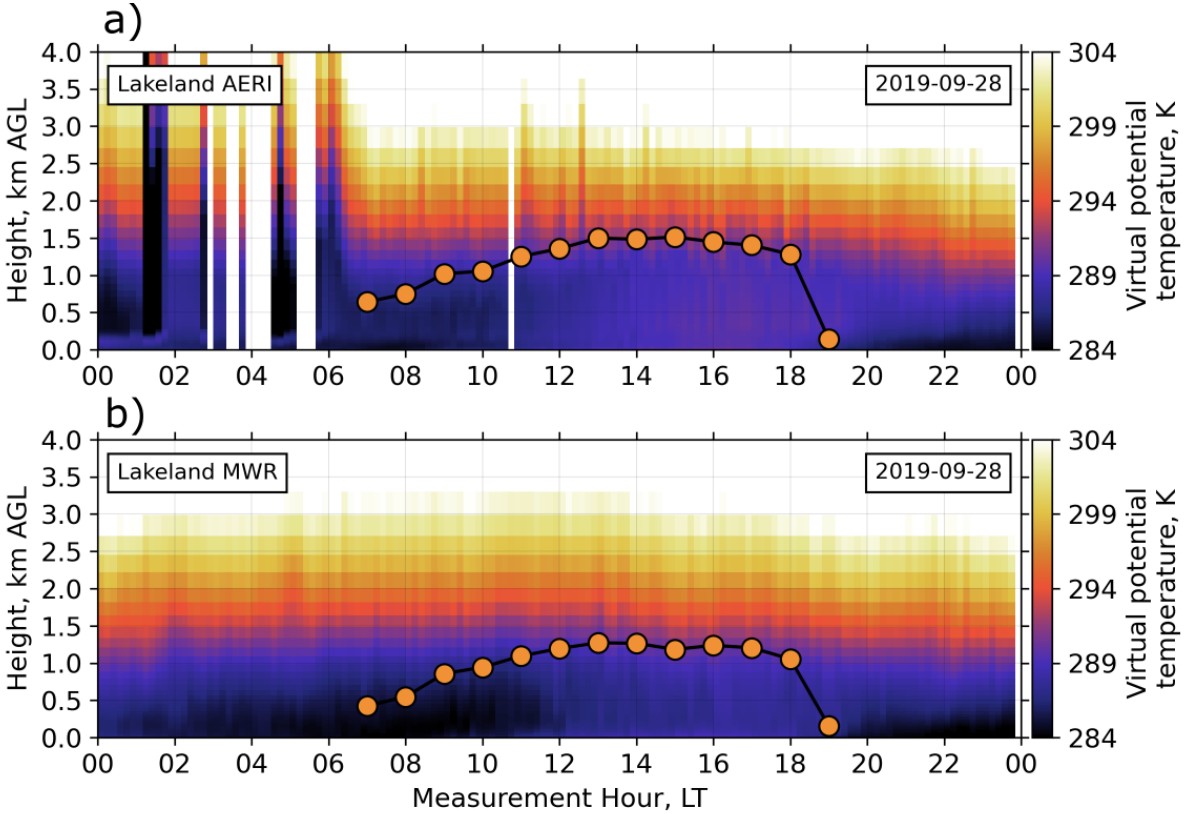

**Figure 3: Time–height cross-section of retrieved virtual potential temperatures for the AERI (a) and for the MWR (b) at the Lakeland site, for day Sep 28, 2019. Overlaid on both panels are PBLHs (orange dots) estimated applying the parcel method to the corresponding hourly-averaged profiles between sunrise (around 07:00 LT) and sunset (around 19:00 LT).**





TROPoe offers the option of incorporating other inputs to improve the thermodynamic profiles retrieved from the AERI and MWR observations, among which are the thermodynamic measurements from active systems such as radio acoustic sounding systems (RASS; Djalalova et al., 2021), and/or thermodynamic profiles from numerical weather prediction models (e.g., Cimini et al., 2011). Other sets of thermodynamic profiles were therefore retrieved when possible (i.e., depending on instrument availability) and archived to the NCAR/EOL data repository (i.e., thermodynamic profiles constructed using only

the passive instrument, using the passive instrument plus RASS, and using the passive instrument plus the Rapid Refresh model, RAP; Benjamin et al. 2016). While including RASS measurements in the TROPoe retrievals can be valuable for improving the accuracy of temperature estimates in the lower part of the atmosphere (where RASS measurements are available), with enhanced accuracy extending above the maximum height of the RASS measurements (Djalalova et al., 2021), its inclusion is not found to significantly impact the PBLH estimates. Similarly, while outputs from a numerical weather

prediction model may outperform traditional retrieval methods for temperature and humidity profiling (Cimini et al., 2011), this inclusion is not found to significantly impact the PBLH estimates. Therefore, the retrieved profiles including RASS and RAP input are not used in this study.

## 2.3 Active Remote Sensing Instruments

Active remote sensing instruments emit a signal and measure the return to extract physical measurements of interest. Several
active ground-based systems deployed during the field campaign were used for this study, including two scanning Doppler lidars deployed in association with the CLAMPS platforms, two CL51s, and two RWPs.

### 2.3.1 CLAMPS multi-instrument

Both CLAMPS multi-instrument platforms deployed for CHEESEHEAD19 were part of a collaborative effort between the NOAA National Severe Storms Laboratory (NSSL) and the University of Oklahoma (OU). The CLAMPS is made up of three
main instruments: a Doppler lidar, a MWR, and an AERI. Together, these instruments allow the CLAMPS to collect high temporal resolution profiles of temperature, moisture, wind, and turbulence information every few minutes or faster depending on the user-configurable scan strategy. The OU-NSSL CLAMPS and the NOAA-NSSL CLAMPS were deployed for CHEESEHEAD19 at Lakeland and Prentice, respectively. Both CLAMPS use Halo Photonics Doppler lidars, which are coherent scanning lidars operating at 1.5-micron wavelength. The CLAMPS at Lakeland uses the Streamline model with
optional upgrades to enable a larger Nyquist limit (±38 m s-1) and increased max range (~10 km) and is equipped with an RPG HATPRO-G4 MWR and an AERI. The CLAMPS at Prentice uses a Streamline XR+ model which has the same capabilities as the Streamline, but with slightly higher laser power and a greater max range (~12 km), and is equipped with a Radiometrics MP3000-A MWR and an AERI, which was unfortunately not operating during CHEESEHEAD19.



Given the CLAMPS structure, this platform provides a unique opportunity to explore multi-instrument value-added products,
such as a fuzzy logic PBLH estimate. In the case of CLAMPS, these value-added products have the benefit of high temporal
resolution, which can be critical for some applications. The fuzzy logic algorithm used here was developed by Smith and Carlin
(2021) and uses a two-step approach wherein actively-sensed measures of turbulence (direct measures of turbulence or mixing
activity) are used in a first-step estimate, which is then refined in a second-step estimate incorporating different indicators of
mixing (measurements suggesting that mixing or turbulence has occurred). This approach is based on the lidar-only algorithm
first developed by Bonin et al. (2018), but additionally leverages multi-instrument inputs (i.e., thermodynamics). Because the
first step almost exclusively depends on the Doppler lidar[1] and the second step effectively refines the initial PBLH estimate,
limits on the ability of the Doppler lidar to penetrate the full depth of the PBL (due to low aerosol loads, cloud layers, etc.) can
limit the ability of the algorithm to successfully estimate the PBLH.

Observations from the CLAMPS at Prentice are shown in Fig. 4 from 28 September 2019. From the Doppler lidar, horizontal
wind speed and direction are retrieved from a 70-degree plan position indicator (PPI) scan using a velocity azimuthal display
technique (Fig. 4a; speed color fill, direction arrows). Vertical velocity observations (positive indicates upward motion; Fig.
4b) are effectively radial velocity measurements from a zenith pointing stare, which is the position the lidar takes and maintains
between each PPI scan. PPIs occur every 15 minutes at the Lakeland site, and every 5 minutes at Prentice. Thermodynamic
profiles (Fig. 4c) of water vapor mixing ratio (color fill) and potential temperature (contour) are retrieved from the respective
passive remote sensors onboard. Hourly estimates of PBLH from the CLAMPS fuzzy logic algorithm are overlaid as orange
circles, with 10-minute estimations plotted as a black curve behind them. Given the high temporal resolution this approach
provides, standard deviation of all 10-minute estimates within a sliding 1-hour wide centered window is also provided
alongside each 10-minute PBLH estimate as a quasi-measure of variability. This standard deviation is displayed in Fig. 4 as
the grey shading along the PBLH curves.

---

[1] The Smith and Carlin (2021) algorithm does include inversion height as detected by the thermodynamic retrieval from CLAMPS as a time-weighted input in the first-step. The time-weighting is a function of local sunrise-sunset time and only allows the inversion height to influence the PBLH estimate during the overnight hours and with a smoothly reducing (increasing) weight during the morning (evening) transition periods.



**Figure 4: (a) Time–height cross-section of horizontal wind speed and direction (fill and arrows, respectively), (b) vertical velocity (positive indicates upward motion) as detected by the Doppler lidar and TROPoe retrieved water vapor mixing ratio (fill), and (c) potential temperature (contour; 285–300 K every 2.5 K) from AERI and MWR observations from the CLAMPS deployed at the Prentice site on 28 September 2019. Overlaid are hourly (orange dots) and 10-min (black curve) PBLH estimates from the CLAMPS fuzzy logic algorithm. The grey fill behind the PBLH estimates is a**






**quasi-measure of variability represented by the standard deviation of all 10-min estimates within a sliding 1-hour wide window centered on each 10-min estimate.**

### 2.3.2 CL51 ceilometer

The backscatter profiles of the CL51 ceilometer measure the profile of attenuated aerosol backscatter, which is correlated with

aerosol concentration and in turn reveals details about the vertical structure of the atmosphere. For instance, the concentration of the scattering aerosols within the mixed layer is expected to be nearly uniform, and higher than in the free troposphere above the mixed layer. Consequently, a method that can be used to estimate the PBLH with this instrument is the so-called "gradient" method that looks for the maximum of the negative gradient of the backscatter profile. Another method, called the "profile fit" method consists of fitting an idealised backscatter profile to the observed range-corrected ceilometer backscatter profile

(Münkel et al., 2007). Vaisala's boundary-layer view (BL-View) proprietary software provides high-temporal resolution (16-s) estimates of the PBLH (i.e., the top of the lowest aerosol layer). Based on the CL51 settings and the configuration of the BL-View software during CHEESEHEAD19, a software-provided merged gradient and profile fit method is used to determine the PBLH. According to the manufacturer BL-View User Guide, this merged method combines the strengths of the two methods, holding the capability to distinguish multiple layers in the profile of the backscatter.

In addition to the high-temporal resolution estimates of the PBLH, the BL-View software provides hourly-mean estimates of the PBLH (shown as the cyan diamonds in Fig. 5). However, the algorithm used to determine the hourly-mean PBLH from these high-temporal resolution estimates is not publicly available, not allowing for much flexibility on the time average of the high-temporal resolution estimates, which is important for this study in order to align the hourly-mean CL51 PBLH over the same time-periods of the other instruments. For this reason, an approach similar to Mues et al. (2017) is used to determine

hourly average estimates of PBLH. This technique establishes a score for each 16-s PBLH estimate within the one-hour period ($\pm$30 min of the hour) and the PBLH exhibiting the highest score is defined as the hourly mean PBLH. In our approach, the score $S(PBLHt_x)$ is calculated for every 16-s PBLH ($PBLHt_x$) within one hour, as:

$$S(PBLHt_x) = \sum_{(PBLHt_x - 200) < PBLH_i < (PBLHt_x + 200)} QC_i \left(1 - \frac{|PBLH_i - PBLHt_x|}{200}\right), \qquad (1)$$

where $PBLH_i$ are the high-frequency PBLH values within the one-hour period which are within a range of 200 m around (i.e.

$\pm$) $PBLHt_x$. Each summation factor for the weight score is scaled by the respective quality control, $QC_i$, index value (provided by the BL-View software) prior to summation (Christopher Münkel, personal communication). The PBLH corresponding to the highest score is reported as the PBLH for the hour.

Close agreement between the hourly estimates derived using Eq. 1 ("QC-scaled approach", orange dots) and the hourly estimates provided by the CL51 BL-View software (cyan diamonds) is demonstrated in Fig. 5. The BL-View software has an

upper limit for PBLH values of 4 km agl and there are some hours when no PBLH value is provided.



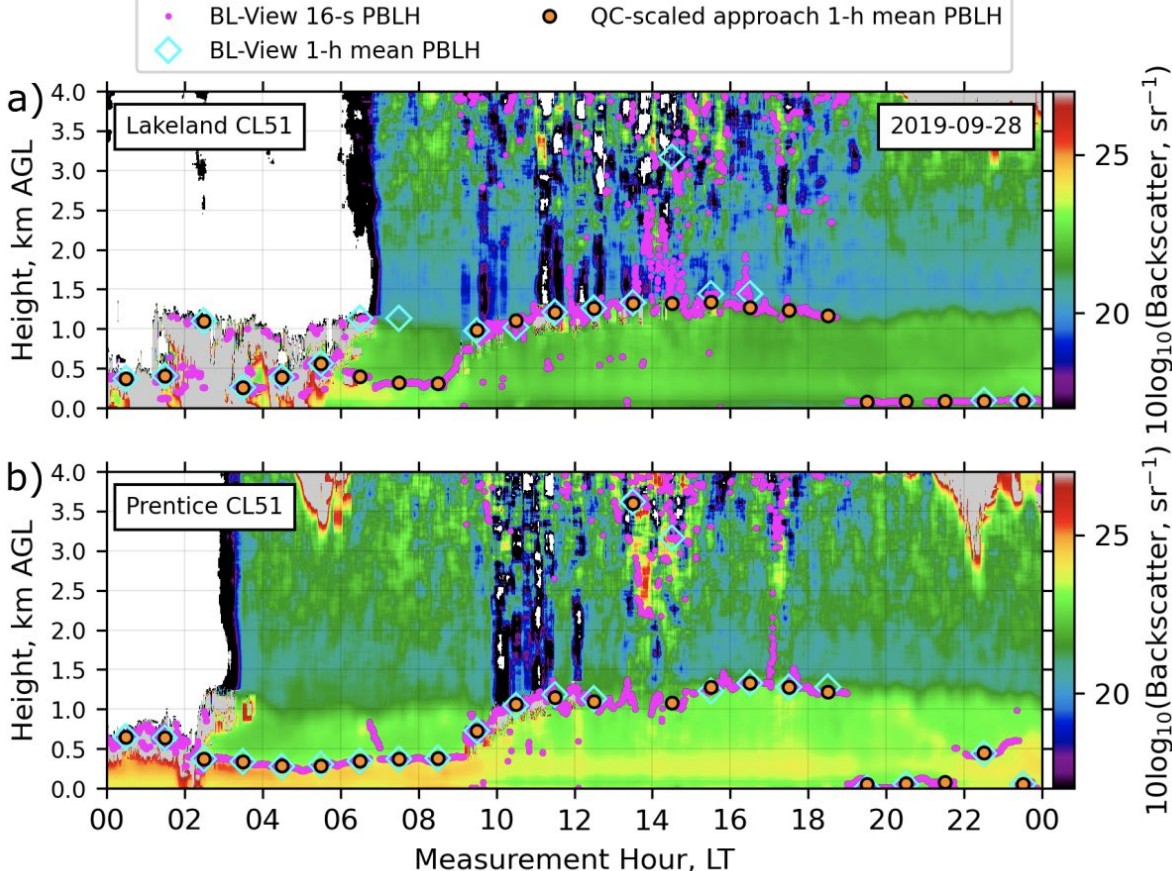

**Figure 5: Time–height cross-section of backscatter profiles at the Lakeland (a) and Prentice (b) sites for 28 Sep 2019. In both panels the magenta dots are the 16-s temporal resolution PBLH estimates by the BL-View software, the cyan diamonds are the corresponding hourly mean estimates from the BL-View software, and the orange dots are the PBLH estimates obtained using the QC-scaled approach.**

The QC-scaled approach is able to reproduce the hourly PBLH BL-View output data with a coefficient of determination ($R^2$) value of 0.87 at Lakeland and an $R^2$ value of 0.95 at Prentice (Fig. 6). Ordinary least-squares regression is used to derive the linear fit between the two hourly-mean PBLH estimates. The root mean square error (RMSE) is found to be equal to 0.28 km at Lakeland and 0.17 km at Prentice and the bias is found to be negligible (i.e., nearly zero) at both sites.



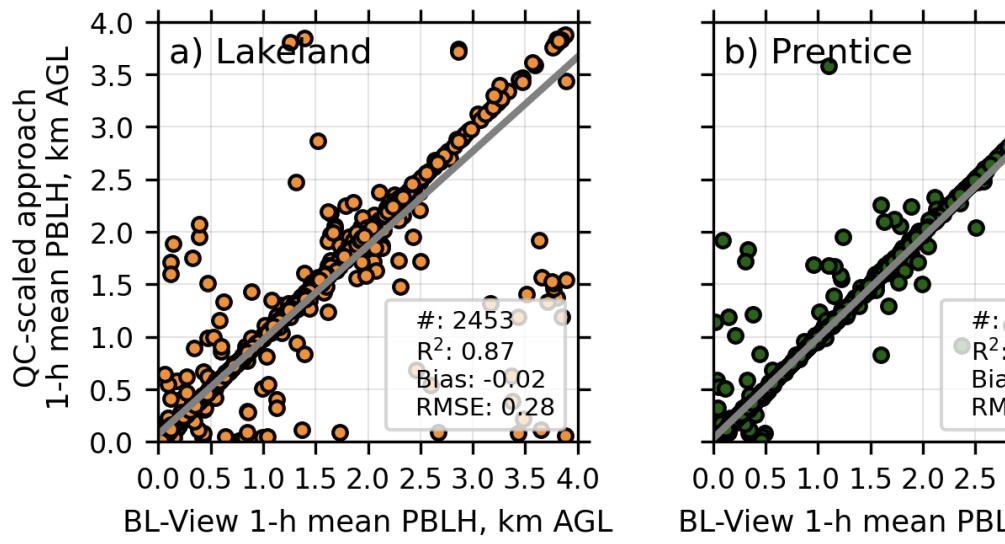

**Figure 6: Scatter plots of PBLH estimates from the CL51 using the BL-View software output (x-axes) and the QC-scaled approach (y-axes) at the Lakeland (a) and Prentice (b) sites over the entire field campaign.**

Despite strong similarity between the BL-View and QC-scaled approach PBLH estimates, some differences do exist. These slight differences are due to other outlier removal methods, cloud filters, and quality control procedures that cannot be replicated in the QC-scaled approach due to the proprietary nature of the BL-View software. However, when these (rare) differences occur (approximately 2% of the 2453 total number of points shown in Fig. 6 for Lakeland, and 1% of the 2507 total number of points at Prentice), the QC-scaled approach appears to provide more reasonable PBLH estimates than the

hourly BL-View output. For example, for hours 06:30, 07:30, 08:30, 16:30 LT at the Lakeland site and hours 14:30 and 15:30 LT at both the Lakeland and Prentice sites (Fig. 5). This holds true when all the days are visually inspected. In addition, because the QC-scaled approach provides flexibility in defining the averaging time period, and therefore allows for averaging time periods to align with other instruments and with the radiosonde launch times, the QC-scaled approach is used to gauge the CL51 PBLH estimation performance.

**2.3.3 RWP**

Two 915-MHz RWP were deployed by the NOAA/Physical Science Laboratory at Lakeland and Prentice. Many studies have shown success in determining the convective boundary-layer depth with RWPs using the information contained in the vertical profile of the SNR (White, 1993; Angevine et al., 1994; Coulter and Holdridge, 1998). In this study, a fuzzy logic-based method (Bianco et al., 2008) is employed that improves the estimation of the PBLH over these original methods by including:

(1) profiles of the variance of vertical velocity, (2) small-scale, radar sub-pulse volume turbulence information from the vertical



profiles of the spectral width of the vertical velocity, and (3) using vertical profiles of the radar-derived SNR. A reliability threshold value is applied to the fuzzy logic-derived score to eliminate PBLH data values with low score values. The automated estimations of the PBLH with the fuzzy logic approach are additionally visually inspected to eliminate suspicious estimations. The use of this last quality control check does not allow for this method to be considered at an operational-level readiness, but
these estimates have been utilized in studies for model validation (Bagley et al., 2017; Bianco et al., 2021) and process understanding (e.g., Bianco et al., 2011; Jeong et al., 2012a, b). The three panel plots provided in Fig. 7 detail the physical quantities measured by the RWP at Lakeland that are used in the fuzzy logic-based approach to discern the PBLH evolution during the daytime (RWP-measured range-corrected SNR, the vertical velocity component of the wind, and the spectral width of the vertical velocity component). Between sunrise and sunset, it is easy to detect the development of the convective boundary
layer, with larger values of range-corrected SNR present at its top (Fig. 7a), strong up- and downdrafts within the PBL during the convective period (positive indicates downward motion; Fig. 7b), and larger values of the spectral width of the vertical velocity component (Fig. 7c), an indication of the turbulent nature of the boundary layer during the convective period. RWPs are not optimal to derive PBLHs during night time, when the PBLH might be lower than the first available measurement from the instrument (about 100m) and the turbulence is weaker.

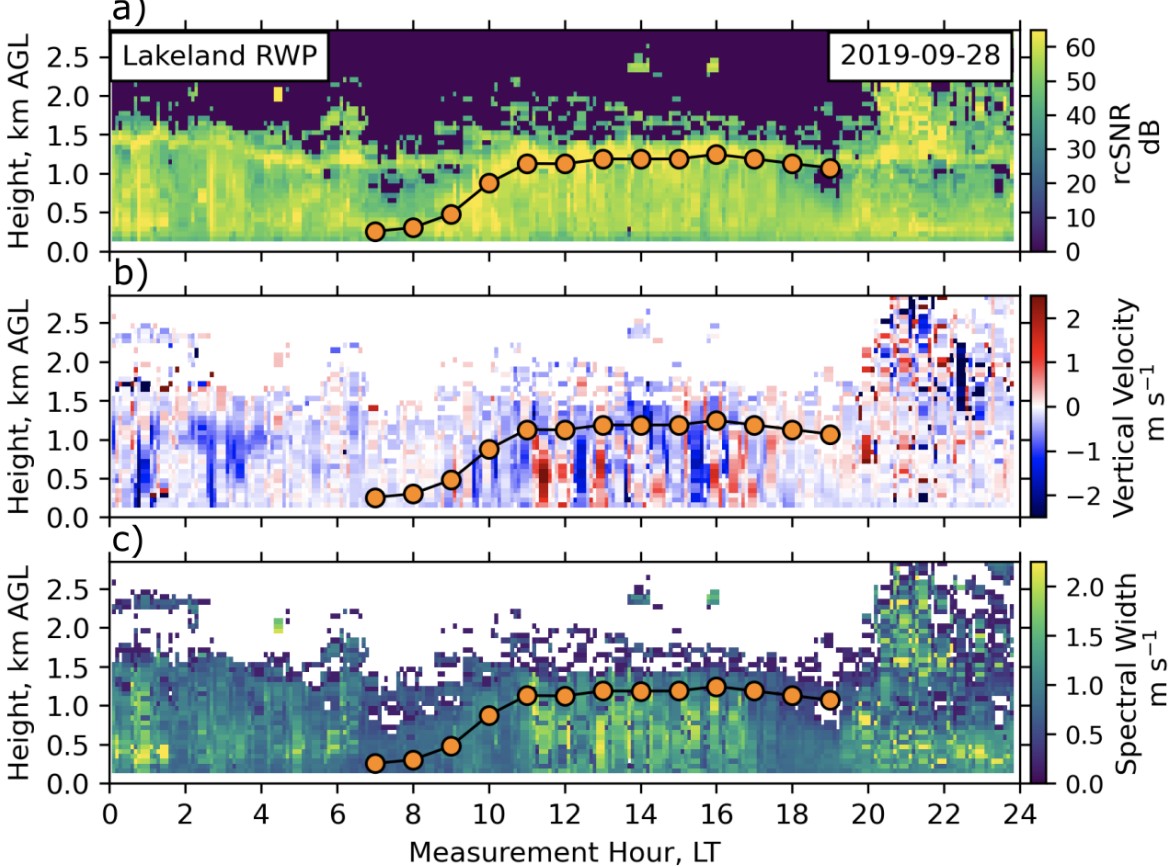





**Figure 7: Time–height cross-section of the RWP measurements used to determine the PBLH estimates at Lakeland for 28 Sep 2019. a) range-corrected SNR; b) vertical velocity (positive indicates downward motion); c) spectral width of the vertical velocity. Overlaid on each of these subplots are the fuzzy logic based hourly-mean PBLH estimates (orange dots) during the daytime.**

**2.3.4 HSRL**

The HSRL deployed at the WLEF site as part of the SPARC instrumentation suite emits a zenith-pointing beam at 532 nm and was active for the duration of CHEESEHEAD19. Data from HSRL were processed into absolutely-calibrated profiles of aerosol backscatter at a temporal and vertical resolution of 30 s and 30 m respectively. The HSRL backscatter coefficient data were visually analysed to subjectively determine the PBLH. Observations from the HSRL are shown in Fig. 8 from 28

September 2019. At the HRSRL site a cleaner air mass with lower backscatter intrudes the PBL after 16:00 LT, in agreement with the CL51 observations at the ISS site (not shown). This is not visible at Lakeland nor Prentice, denoting a spatial variability in PBLH in the afternoon for this day.

The HSRL dataset was visually evaluated to detect the sharp gradient in aerosol backscatter at the top of the PBL, to serve as an independent "expert" dataset to help provide context during two of the IOPs.

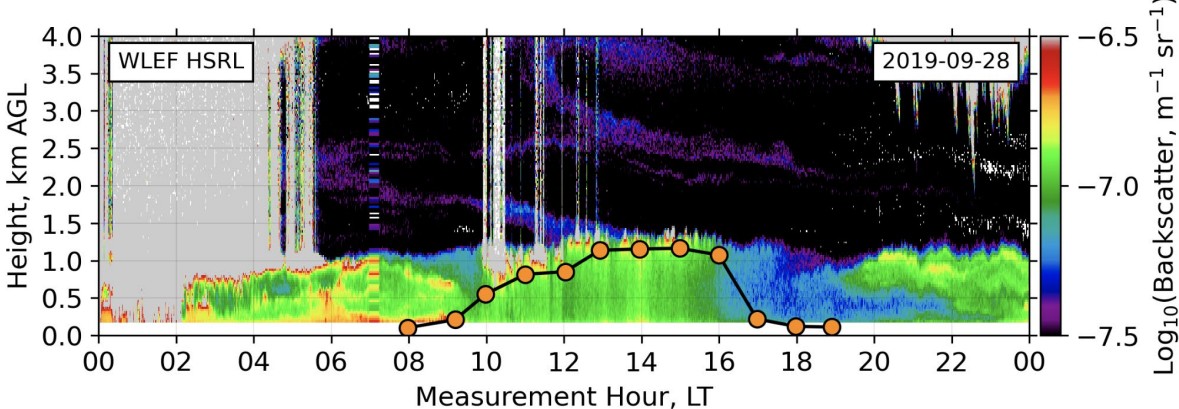

**Figure 8: Time–height cross-section of the HSRL aerosol backscatter measurements at the WLEF site that were used to subjectively determine the PBLH estimates for 28 Sep 2019. Overlaid are the subjectively determined hourly-mean PBLH estimates (orange dots) during the daytime.**

PBLHs are derived using data measured by all the instruments between ±30 minutes of the hour to extract hourly averaged values. For example, data measured between 12:30 LT and 13:30 LT are used to determine the 13:00 LT hourly averaged estimate of the PBLH values over that hour, and are compared to the PBLH estimate from the 13:00 LT radiosonde launches.



## 2.4 RadSys Stations

Two RadSys stations were deployed at Lakeland and Prentice (data available from Riihimaki, et al., 2020a, b) providing cloud (e.g. cloud fraction) and radiation information. The cloud fraction provided by the RadSys instrumentation is a derived variable calculated from broadband shortwave radiometer measurements using algorithms established by Long et al. (2000; 2006). The hour window used to determine the cloud fraction is the same temporal window ($\pm$30 min of the hour) used to determine the PBLH estimates.

## 405 3 Evaluation and Characterization of PBLHs

Table 2 provides a summary of the instruments and the methods used to determine PBLH estimates from these instruments. While the MWR and AERI PBLH estimates are based on the same parcel method as the radiosondes, the other methods are more related to the identification of gradients in turbulence or backscatter.

| Instrument System | PBLH Estimation Technique |
|---|---|
| Radiosondes | Parcel Method |
| AERI | Parcel Method |
| MWR | Parcel Method |
| CLAMPS multi-instrument | Fuzzy Logic |
| CL51 | QC-scaled approach (adapted from Mues et al., 2017) |
| RWP | Fuzzy Logic |
| HSRL | Visual Inspection - Expert Opinion |


**Table 2: Instrument and methods used to discern PBLHs.**

As a first step, PBLH estimates from the RWPs, the CL51s, and the MWRs at the Lakeland and Prentice sites are evaluated against radiosonde-based PBLH estimates for the entire campaign period. Comparison was limited to these three instruments 415 because the RWPs and the CL51s were deployed at these sites for the entire field campaign, providing a large dataset to analyze, while the MWRs were deployed 34 (Prentice) and 35 (Lakeland) days after the start of CHEESEHEAD19 (see Table 1). Some of the other instruments were moved between sites during the campaign, were deployed for a much shorter period of time, or experienced failures during the experiment. Therefore, these instruments were not included in the bulk analyses. In a second step, a validation on a more limited dataset, over two of the IOPs, is performed that includes all of the available 420 instruments.





## 3.1 Multi-Month Time Period Analysis of the RWP, CL51, and MWR PBLH Estimates

Figure 9 evaluates the PBLH estimates from the RWPs, the CL51s, and the MWRs deployed at Lakeland (Fig. 9a, b, c, respectively), and Prentice (Fig. 9d, e, f, respectively) against the radiosondes estimates. Using the cloud and radiation information from the RadSys stations deployed at Lakeland and Prentice, comparisons as a function of the cloud fraction (a

value of zero indicating clear skies) were additionally made. All available estimations from each instrument are plotted.

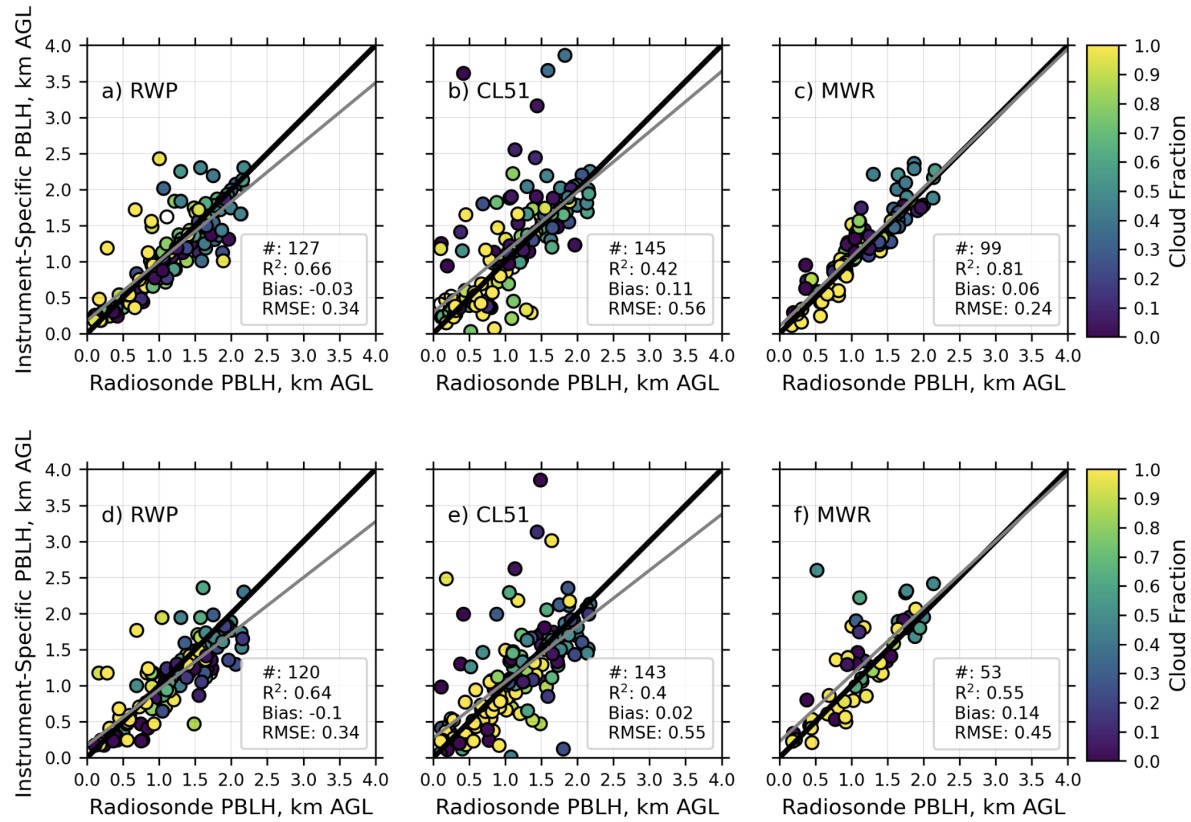

**Figure 9: Comparisons of the PBLHs (comparisons at the Lakeland site are provided in panels a, b, and c, and those**
**at the Prentice site in panels d, e, and f) from the parcel method applied to the radiosonde virtual potential temperature profiles (x-axes) versus those estimated (y-axes) by the RWP fuzzy logic method (a and d), the CL51 QC-scaled approach (b and e), and the parcel method applied to the MWR virtual potential temperature profiles (c and f). Comparisons are categorized by the cloud fraction coverage (colorbar).**

Both MWRs deployed at the Lakeland site are used in this analysis to maximise the number of available estimations from this instrument. However, since the MWRs were deployed later (compared with the RWP and CL51), fewer points are available for the MWR statistics. PBLH estimates are additionally time-matched so comparisons are made when PBLH estimates are



available from all three instruments at each site and the statistical analysis was repeated. Results from both approaches are presented in Table 3.

| Site | | Instrument System | # of points | $R^2$ | Bias (m) | RMSE (m) |
|---|---|---|---|---|---|---|
| Lakeland | All points | RWP | 127 | 0.66 | -30 | 340 |
| | | CL51 | 145 | 0.42 | 110 | 560 |
| | | MWR | 99 | 0.81 | 60 | 240 |
| | Time-matched points | RWP | 62 | 0.64 | -60 | 310 |
| | | CL51 | 62 | 0.23 | 120 | 620 |
| | | MWR | 62 | 0.80 | 80 | 240 |
| Prentice | All points | RWP | 120 | 0.64 | -100 | 340 |
| | | CL51 | 143 | 0.4 | 20 | 550 |
| | | MWR | 53 | 0.55 | 140 | 450 |
| | Time-matched points | RWP | 42 | 0.77 | -80 | 260 |
| | | CL51 | 42 | 0.13 | 70 | 760 |
| | | MWR | 42 | 0.75 | 70 | 310 |

**Table 3: Statistical comparison of PBLH estimated by the RWPs, CL51s, and MWRs compared with the radiosonde estimates during the entire CHEESEHEAD19 campaign for the Lakeland and Prentice sites. Results are presented for all available data points (upper portion of the section relative to each site), and for the time-matched data points (bottom portion of the section relative to each site).**

Comparing the time-matched statistics at the Lakeland site, the MWR performs better than the RWP, whose methodology does not include any thermodynamic information, while they behave similarly at the Prentice site. The CL51 estimates, also not including any thermodynamic information, tend to have a larger spread in general compared with both the RWP and MWR.



A way to improve PBLH estimates for all instruments, but particularly for the CL51, could be to check for temporal consistency in the hourly PBLH estimates; filtering out large jumps that can arise for example in the presence of elevated aerosol layers (see case study 1 in Sect 3.3).

In Fig. 10 the error in PBLH estimates between the RWP, the CL51, the MWR, and the radiosondes is presented as a function of the time of the day for the two sites, using all available estimations from each instrument. Larger errors are shown for the CL51 compared with the other two instruments for nearly all the time examined, but particularly during the morning transition period.

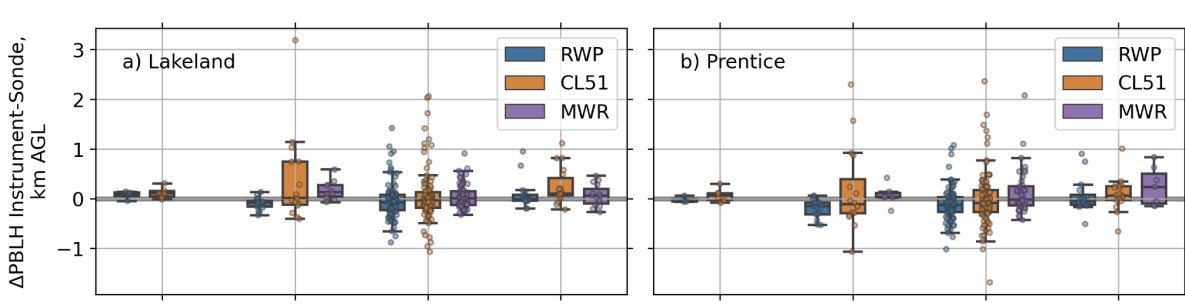

**Figure 10: Statistical comparison of the PBLHs from the parcel method applied to the radiosonde virtual potential temperature profiles versus those estimated by the RWP fuzzy logic method, the CL51 QC-scaled approach, and by the parcel method applied to the MWR, at the Lakeland (a) and Prentice (b) sites, as a function of the hour of the day.**

The CL51 frequently misidentifies elevated aerosols layers as the PBLH, particularly when the gradients are not clearly pronounced, as for instance at hour 09:00 LT (Fig. 10 a, b). The ceilometer method also struggles with detecting the decay of the planetary boundary layer in the late afternoon, as the instrument continues to mistake the height of the residual boundary layer for the PBLH (i.e., at hour 17:00 LT), leading to a positive bias at both the Lakeland and Prentice sites (Fig. 10 a, b). During the decaying phase of the boundary layer (17:00 LT), the RWP errors are the smallest and centered around zero at both sites (likely because the RWP can detect the decay of turbulence during the evening transition), while the MWR errors are centered around zero at the Lakeland site, but positive at the Prentice site due to the inability of the MWR to accurately resolve a sharp inversion at the top of the convective boundary layer.

To conclude the multi-month time period analysis, evaluation of the PBLH estimates were made relative to the RadSys cloud fraction information (Fig. 11). Comparisons were divided into clear-sky days (defined by RadSys cloud fraction values less than 0.1, Fig. 11a, b, c), extremely cloudy days (defined by RadSys cloud fraction values larger than 0.9, Fig. 11g, h, i), and for the cases in between these two cloud fraction extremes (Fig. 11d, e, f). For the clear sky days, the coefficient of determination in the CL51-radiosondes comparison is lower ($R^2 = 0.28$, Fig. 11b) than both the RWP ($R^2 = 0.84$, Fig. 11a) and





the MWR ($R^2$ = 0.82, Fig. 11c). RMSE and bias are also larger for the CL51. This is likely due to the above-mentioned

475 limitation of the CL51 at identifying the PBLH when aerosol gradients are not strongly pronounced (Fig. 10 a, b).

For the days with cloud fraction between 0.1 and 0.9, the coefficient of determination between the radiosonde PBLH and both

the RWP ($R^2$ = 0.66, Fig. 11d) and the MWR ($R^2$ = 0.54, Fig. 11 f) PBLH estimates are not as good as for clear-sky days.

Despite this, their performance is still better than that of the CL51 ($R^2$ = 0.48, Fig. 11 e). The same is also true for RMSE. In

both Fig. 11b and e the CL51 detects several elevated PBLHs when the radiosonde values are shallower.

480 Finally, the heights of cloud-topped PBLs are more difficult to measure and overall more difficult to understand. For the RWP,

an overall positive bias is observed compared with the radiosonde PBLH estimates, agreeing with Grimsdell and Angevine

(1998) that the peak in the RWP radar return tends to be slightly higher than the cloud base due to both the increased turbulence

within the cloud and the effect of sharp moisture gradients at the cloud edges. With the exception of the few outliers in the

upper portion of Fig. 11h, the CL51 retrieved PBLHs are generally lower than the radiosonde estimates, indicating selection

485 of the PBLH at the base of the detected cloud. The MWR statistics in cloudy conditions (Fig. 11h), however, are nearly as

good compared to during more clear sky conditions (Fig. 11c). This is probably due to the low opacity of the cloud at

microwave wavelengths and thus the relative insensitivity of the retrieved thermodynamic profile to cloud liquid water.

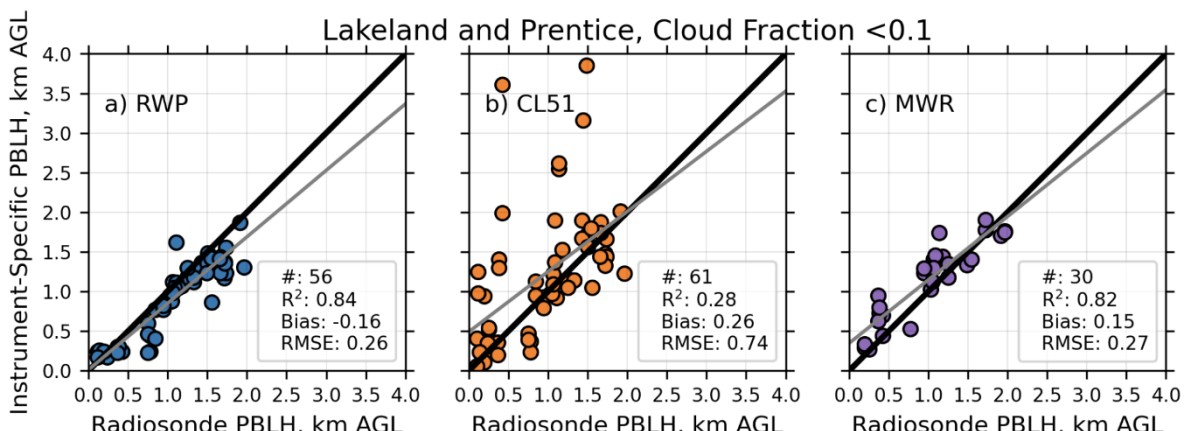





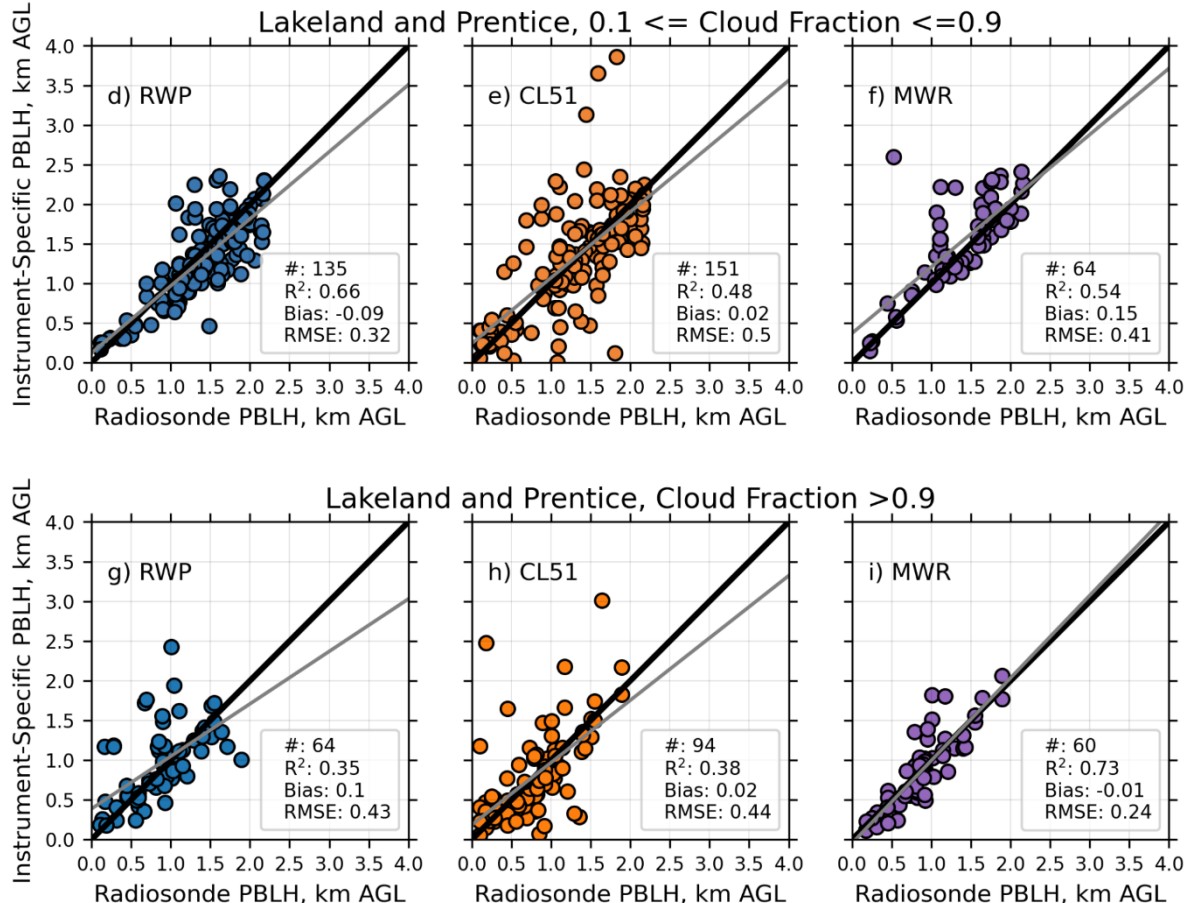

**Figure 11: Same as in Fig. 9 at the Lakeland and Prentice sites but divided for the times with RadSys cloud fraction values less than 0.1 (a, b, c), for the times with RadSys cloud fraction values between 0.1 and 0.9 (d, e, f), and for the times with RadSys cloud fraction values larger than 0.9 (g, h, i).**

## 3.2 Multi-Instrument Validation (Summer-IOP and Fall-IOP)

The second part of the comparison concentrates on two limited time periods, the period between 19 and 24 August (referred to as Summer-IOP) when all the instruments were available with exception of the CLAMPS (i.e., the RWP, the CL51, the MWR, the AERI and the HSRL), and the period between 19 September and 5 October 2019 (referred to as Fall-IOP) when the CLAMPS were also available. Using the information from the RadSys station, a similar cloud fraction analysis to that in Section 3.1 is performed on these two IOPs.

All instruments for the Summer-IOP are from the Lakeland site, except for the ISS-based radiosondes, and both the HSRL and the AERI located at the WLEF site.





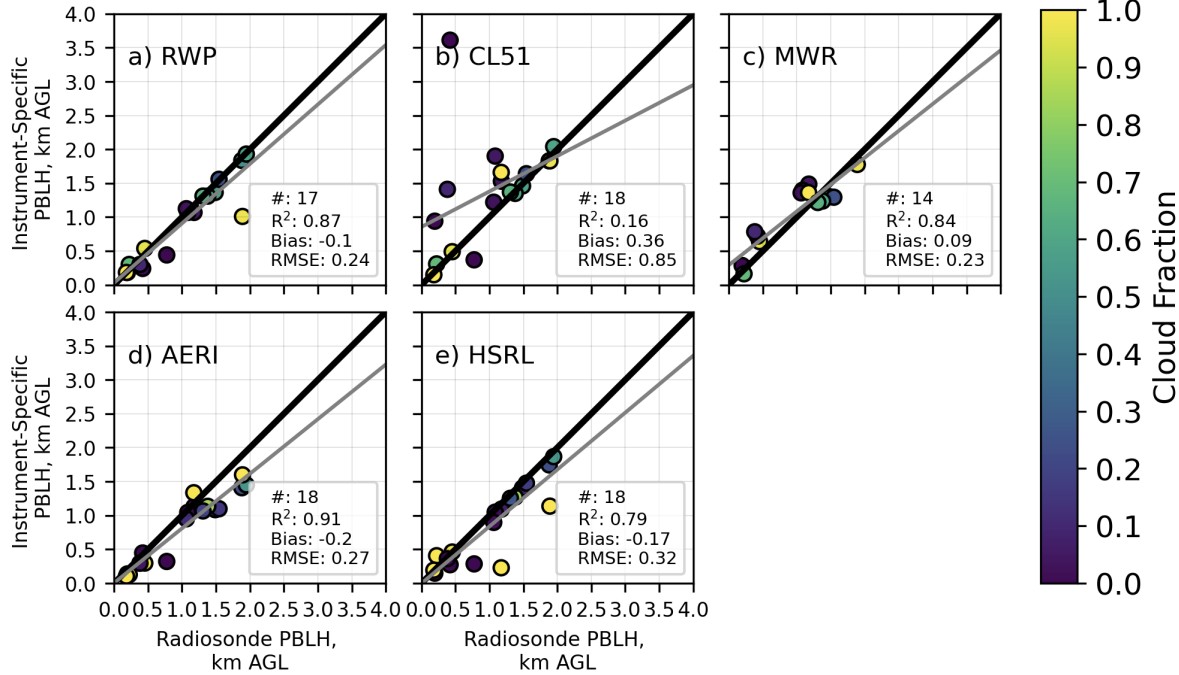

**Figure 12: Comparison of the PBLHs by the parcel method applied to the radiosonde virtual potential temperature profiles (x-axes) versus those (y-axis) estimated (a) by the RWP fuzzy logic method; (b) by the CL51 QC-scaled approach; (c, d, respectively) by the parcel method applied to the MWR and AERI virtual potential temperature profiles; and (e) by the expert opinion to the HSRL observations, for the Summer-IOP period. The color coding shows the site-respective cloud fraction values.**

For the Fall-IOP analysis, the RWP, CL51, MWR, AERI, HSRL, and the CLAMPS are all available. All instruments used in the Fall-IOP are from the Lakeland site, except for ISS-based radiosondes, the WLEF-based HSRL, and the CLAMPS located at Prentice. Although a CLAMPS was also available at the Lakeland site, it had limited height coverage compared to the one at Prentice, never providing PBLH estimates above 1 km agl (likely due to low aerosol concentrations and instrument low signal power). For this reason, the Fall-IOP analysis used the Prentice-based CLAMPS.





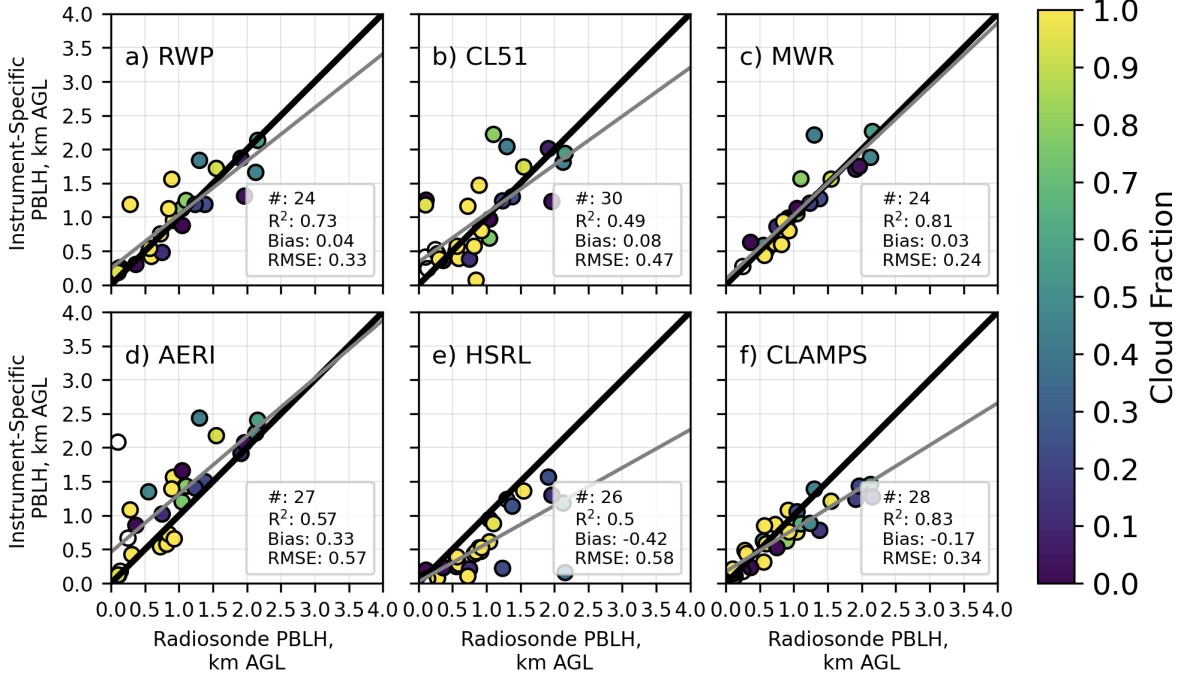

**Figure 13: As in Fig. 12, but for the period Fall-IOP and with CLAMPS included. White dots are relative to missing data from the RadSys.**

Similar to the approach for the multi-month time period analysis, PBLH estimates were time-matched to identify times when height estimates were derived from each instrument. Statistical analysis with and without this time-matching procedure are provided in Table 4.

| IOP | | Instrument System | Deployment Locations | # of points | $R^2$ | Bias (m) | RMSE (m) |
|---|---|---|---|---|---|---|---|
| | | RWP | Lakeland | 17 | 0.87 | -100 | 240 |
| | | CL51 | Lakeland | 18 | 0.16 | 360 | 850 |
| **Summer-IOP** | **All points** | MWR | Lakeland | 14 | 0.84 | 60 | 220 |
| | | AERI | WLEF | 18 | 0.91 | -200 | 270 |
| | | HSRL | WLEF | 18 | 0.79 | -170 | 320 |





| | | | | | | | |
|---|---|---|---|---|---|---|---|
| | **Time-matched points** | RWP | Lakeland | 13 | 0.81 | -90 | 260 |
| | | CL51 | Lakeland | 13 | 0.04 | 500 | 990 |
| | | MWR | Lakeland | 13 | 0.84 | 60 | 230 |
| | | AERI | WLEF | 13 | 0.95 | -170 | 220 |
| | | HSRL | WLEF | 13 | 0.87 | -110 | 230 |
| **Fall-IOP** | **All points** | RWP | Lakeland | 24 | 0.73 | 40 | 330 |
| | | CL51 | Lakeland | 30 | 0.49 | 80 | 470 |
| | | MWR | Lakeland | 24 | 0.81 | 30 | 240 |
| | | AERI | Lakeland | 27 | 0.57 | 330 | 570 |
| | | HSRL | WLEF | 26 | 0.5 | -420 | 580 |
| | | CLAMPS multi-instrument | Prentice | 28 | 0.83 | -170 | 340 |
| | **Time-matched points** | RWP | Lakeland | 17 | 0.69 | -20 | 310 |
| | | CL51 | Lakeland | 17 | 0.48 | 50 | 440 |
| | | MWR | Lakeland | 17 | 0.75 | 70 | 280 |
| | | AERI | Lakeland | 17 | 0.66 | 360 | 480 |
| | | HSRL | WLEF | 17 | 0.35 | -510 | 680 |
| | | CLAMPS multi-instrument | Prentice | 17 | 0.71 | -280 | 410 |

**Table 4: Statistical comparison of PBLH estimated by the RWP, CL51, MWR, AERI, HSRL, and CLAMPS compared with the radiosonde-based PBLH estimates for both the Summer-IOP and Fall-IOP analysis. Results are presented for all available data points (upper portion of the section relative to each IOPs), and for the time-matched data points (bottom portion of the section relative to IOPs).**



Figures 12 and 13, and the results presented in Table 4, demonstrate that for both the Summer-IOP and Fall-IOP analyses, the RWP, MWR, and AERI perform reasonably well in resolving PBLH.

Fig. 14 presents the statistical comparison between the derived PBLHs and the radiosonde estimates as a function of the hour of the day for the Summer-IOP (Fig. 14a) and for the Fall-IOP (Fig. 14b). CL51 results are consistent with those found over the multi-month time period analysis, having less skill than the RWP and MWR instruments, and for the two IOPs also less

skill than the AERI, especially for the morning and late afternoon transition periods (Fig. 14). This indicates that more refined methods are needed to derive accurate PBLH values from ceilometers. Although based on a limited number of cases, the CLAMPS does well at resolving PBLH for the Fall-IOP analysis (Fig. 13f) with the average negative bias mostly due to the negative bias found at 17:00 LT (Fig. 14b). This negative bias is probably due to the low aerosol concentrations in these midday environments, which limits the vertical coverage of the CLAMPS scanning Doppler lidar (no PBLH estimates above 1.5 km

agl are provided by the CLAMPS, Fig. 13f). As mentioned before, the vertical coverage of the CLAMPS located at Lakeland was lower than the one at Prentice resulting in worse statistics (not shown). The RWP exhibits a bias close to zero, consistent between the two IOPs and independent of the time of day examined (blue bars in Fig. 14a, b). The RWP PBLH agreement with the radiosonde estimates indicates that the mechanism used in the fuzzy logic approach (determining the turbulent layer in contact with the surface) for PBLH estimation is physically consistent with the parcel method approach (determining the

height of the well-mixed layer). The CL51 errors (orange bars in Fig. 14a, b) are less consistent with the time of day and IOP, but in general the errors are larger than the RWP. The MWR biases (purple bars in Fig. 14a, b) are close to zero and the error value is small regardless of the time of day, particularly for the Fall-IOP. The AERI errors (maroon bars) are also small but show a slightly negative bias for the midday and evening transition hours of the Summer-IOP (Fig. 14a), and a positive bias with larger errors for the Fall-IOP period (Fig. 14b). However, the AERI is very sensitive to clouds (e.g., Turner 2007) and the

differences between the two IOPs could be related to the markedly different cloud conditions experienced during the respective IOP (Sedlar et al., 2021). Finally, the expert estimation on the HSRL observations (pink bars) presents very accurate results for the Summer-IOP (Fig. 14a) analysis, but negative biases and larger errors for the Fall-IOP analysis (Fig. 14b). Similar to the AERI results, this could be due to the increased amount of stratiform cloud cover during the Fall-IOP period. This speaks to the fact that the estimation of the PBLH is a difficult task and while it can be easily identifiable on some days, it can be

ambiguous on other days. The Fall-IOP has more cloud coverage (50% on average) compared with the Summer-IOP (36% on average), making the PBLH estimation more challenging not only for expert estimation, but also for all other instruments and the corresponding estimation techniques. The exception to this is the CL51, which performed better in the cloudier Fall-IOP compared to its performance during the Summer-IOP.





**Figure 14: As in Fig. 10, but for all instruments involved in the (a) Summer-IOP and (b) Fall-IOP analysis.**

### 3.3 Multi-Instrument Limitations – Case Studies

Two case studies are investigated to understand in more detail the behaviour of the different instruments in different cloud-covered conditions. Additional plots demonstrating the individual instrument measurements for both case studies are provided
in the Supplemental Material section.

The first day under consideration (19 Aug 2019) is a nearly clear-sky day with no clouds in the boundary layer. For this day, the CLAMPS was not available. Figure 15 provides the daily PBLH estimations from the available instruments (Fig. 15a) and from the radiosondes launched at 09:15, 13:00, and 16:45 LT. The three cloud-base heights (CBH1, CBH2, and CBH3) and the cloud-base fraction (CBF - the fraction of the time within a one-hour window that the ceilometer measures cloud base
below either 3 km or 13 km agl) derived from the CL51 are presented in Fig. 15b. The vertical profiles of virtual potential



temperature (Fig. 15c, d, and e) measured by the radiosonde highlight a well-mixed layer topped by a distinct inversion up to a few hundred meters agl at 09:15 LT. The inversion then rises to approximately 1.2 km and 1.1 km agl at 13:00 LT at 16:45 LT, respectively. The vertical profiles of virtual potential temperature from the AERI and MWR are smoother (as expected from passive instruments) compared to those from the radiosondes; nevertheless, they both correctly represent the diurnal

growth of the PBLH. In particular, the AERI reproduces the 09:00 LT virtual potential vertical profile quite well. In general, most of the instruments reasonably resolve the diurnal cycle of the PBLH. The RWP, AERI, and HSRL agree well with the radiosonde PBLH estimates, while the MWR profile of virtual potential temperature, with its smoother characteristics due to its relatively coarse vertical resolution (compared to the AERI, see Blumberg et al. 2015), places the PBLH estimates slightly higher than the other instruments. Estimates from the CL51 are higher than those from the other instruments because this

instrument detects elevated aerosol layers as the PBLH, perhaps from smoke advected from wildfires on the west coast of North America. These elevated layers are also visible in the HSRL observations, as evident in Figs. S2 (for the CL51) and S5 (for the HSRL) of the Supplemental Material.




**Figure 15: Case study Aug 19, 2019. a) Time series of PBLH estimates from the different instruments between sunrise and sunset. b) Cloud-base height estimates (CBH 1, CBH 2, and CBH 3) and cloud-base fraction (CBF) from the CL51 at Lakeland. c), d), e) Profiles of virtual potential temperature at the radiosonde launch times (09:15, 13:00, and 16:45 LT) from the radiosondes, AERI, and MWR, with colored vertical dashed lines representing application of the parcel**
**method to the respective instrument profile. PBLH estimates at the time of the radiosondes from the different instruments are also denoted by the colored bars in panels e-f.**

The second day investigated (23 Sep 2019) exhibited stratiform clouds in the morning with multi-level cumulus clouds developing in the afternoon (Fig. 16b). The presence of various cloud structures during this day demonstrates the complexity
of PBLH estimation depending upon the boundary layer conditions. All instruments were available on this day, although AERI





profiles were missing before noon. PBLH estimates from the available instruments (Fig. 16a) and from the radiosondes (launched at 09:13, 12:59, and 16:45 LT) on this day are provided in Figure 16.

The PBLH starts increasing early in the morning in conjunction with the vertically developing cloud base. During the morning hours, the remote sensing instruments demonstrate agreement between themselves and the radiosondes, placing the PBLH near
the base of the stratiform clouds, but in the afternoon larger differences emerge. After approximately 12:30 LT, the cloud coverage becomes more scattered and cloud base height is more variable. At 13:00 LT, the PBL is well mixed up to around 1.5 km agl, but the radiosonde places the PBLH somewhere in the cloud layer near 2.1 km agl. After 13:00 LT, PBLHs detected with the RWP and CL51 are at almost 2.5 km agl, higher than the estimates from the other instruments. This high bias is likely caused by the elevated cloud base identified around 2.3 km agl (Fig. 16b). Since radio soundings are not available between
14:00 and 16:00 LT, when the differences between the instrument estimates are particularly large, it cannot be determined whether or not these clouds are connected to the well-mixed layer. Nevertheless, because turbulence remains well developed up to that height (as visible from the spectral width of the RWP vertical velocity, Fig. S6c of the Supplemental Material), the RWP defines the PBLH at the radar return peak found slightly higher than the cloud base at this time. Similarly, the CL51 places the PBLH higher than the other instruments after 13:00 LT, where it finds the maximum of the negative gradient of the
backscatter profile (see Fig. S7 of the Supplemental Material). Throughout the day the HSRL detects a lower PBLH compared to the other instruments, which is close to the lowest cloud base height up to 13:00 LT. Note, the HSRL is not at the Lakeland site, but at the WLEF site, where the lowest cloud base is less scattered than at the Lakeland site (see Fig. S10 of the Supplemental Material). MWR and AERI estimates agree well with those from the radiosondes, although the shapes of their virtual temperature profiles are different and smoother compared to the radiosondes (Fig. 16c, d, and e). For this single day
case, the CLAMPS approach resolves the morning development of the PBL quite well up to hour 13:00 LT, but it does not identify PBLHs higher than 1.5 km agl during the afternoon hours (and in general for the entire diurnal period, as noted in Section 3.2 and Fig. 13f.).





**Figure 16: As in Fig. 15, but for Sep 23, 2019, and with CLAMPS included.**

## 4 Summary and Concluding Remarks

In this study, data collected by both active and passive ground-based remote sensing instruments deployed during the CHEESEHEAD19 field campaign (summer of 2019, northern Wisconsin, U.S.A) are used to estimate the height of the daytime planetary boundary layer, and their values are compared against independent PBLH estimates obtained from radiosondes launched as part of the field campaign. To retrieve PBLH from the thermodynamic profiles of radiosondes, MWR, and AERI the parcel method was used, while the methods used on RWP, CL51, HSRL, and CLAMPS are more related to the identification of gradients in turbulence or backscatter. The impact of boundary layer clouds on boundary-layer depth is also investigated.



For this dataset, results show that:

- RWPs are suitable to be used in estimating the PBLH with a small bias during the entire daily period. The RWP was able to also correctly detect the decaying phase of the PBLH, but demonstrated decreased performances during cloudy conditions when the bias is found to be positive due to the increased turbulence measured by the instrument within the cloud.

- CL51 PBLH estimates based on the high-temporal resolution PBLH estimates provided by the manufacturer software capture the PBLH better during times of overcast cloudiness and can generally capture the convective PBLH when the PBL is well-mixed during mid-day. However, the CL51 retrieval method used here is challenged in clear-sky periods when strong upper level aerosol gradients are present, particularly in the morning and evening transition periods when residual layers exist above the well-mixed layer. This weakness in the method negatively impacts the CL51 statistical

performance.

- MWRs are found to reasonably estimate the PBLH when the parcel method definition is applied to the retrieved profiles of virtual potential temperature. The performance of MWR with this definition is good across the range of cloudy conditions.

- AERIs, although also passive instruments, have higher vertical resolution in their retrieved thermodynamic profiles

compared to MWR. They perform very well in conditions with fewer clouds, with their performance decreasing in cloudy conditions as clouds are opaque to infrared transmission. In this case, the AERI PBLH estimates have a positive bias over the entire daily cycle.

- HSRL observations evaluated by an expert also present very accurate results for the less cloudy data period, with larger errors for the period with increased amount of clouds, which is a reminder of how the PBLH estimation can be easy on

some days and ambiguous on others, even to an expert eye.

- CLAMPS platforms prove to be a valuable possibility for PBLH estimates. Nevertheless, the limited vertical coverage of these platforms due to limits on the ability of the Doppler lidar to penetrate the full depth of the PBL (for instance because of low aerosol concentrations) can be a limiting factor on their PBLH estimation skill.

Another important result of this study is that although some of the instruments analyzed in this study might have the advantage

of using the parcel method to estimate the PBLH (i.e. the same method was applied to the radiosondes to establish a validation dataset), instruments that do not rely on thermodynamic information, such as the RWPs and the HSRL, are in relatively agreement with the radiosonde PBLH estimates.

The results finally show that all of the instruments used in this study are capable of providing reasonable PBLH estimates in most circumstances. However, each of them has weaknesses during certain conditions that should be kept under consideration

when using them for the evaluation of numerical weather prediction models.



## Data Availability

The complete datasets of observations used in this study (as well as the model runs) are freely available for general use through the National Center for Atmospheric Research (NCAR) Earth Observatory Laboratory (EOL) data repository (www.eol.ucar.edu/field_projects/cheesehead).

## Author contribution

JD and BA completed the primary data analysis, JD and LB prepared the manuscript with contributions from all co-authors.

## Acknowledgements

We thank all the people involved in CHEESEHEAD19 for site selection, leases, instrument deployment and maintenance, data collection, and data quality control. Funding for this work was provided by the NOAA Atmospheric Science for Renewable
Energy (ASRE) program, and in part by the NOAA Cooperative Agreement with CIRES, NA17OAR4320101. Data provided by NCAR/EOL under the sponsorship of the National Science Foundation. https://data.eol.ucar.edu/.

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
