# Peer review of "Evaluating convective planetary boundary-layer height estimations resolved by both active and passive remote sensing instruments during the CHEESEHEAD19 field campaign"

_Atmospheric Measurement Techniques, 2021_

## Author Comment (AC1)

**Review of Evaluating daytime planetary boundary-layer height estimations resolved by both active and passive remote sensing instruments during the CHEESEHEAD19 field campaign**

Anonymous Referee #1

Referee comment on "Evaluating daytime planetary boundary-layer height estimations resolved by both active and passive remote sensing instruments during the CHEESEHEAD19 field campaign" by James B. Duncan Jr. et al., Atmos. Meas. Tech. Discuss., https://doi.org/10.5194/amt-2021-363-RC1, 2022

Within this manuscript, the authors compare estimates of the convective planetary boundary layer height (PBLH) made from several different remote sensors with radiosonde derived estimates of the PBLH. The authors discuss key differences in the ability of each remote sensor (or system) to accurately ascertain the PBLH, using the radiosonde estimates as truth. When significant differences are apparent, the authors discuss possible reasons appropriately. This includes a statistical comparison as well as a few case days that were examined in closer detail. Overall, this manuscript is fairly well written and of interest to the Atmospheric Measurement Techniques reader community. While largely developed PBLH algorithms are used, the analysis presented herein further assess the strengths and limitations of each sensor and associated method to determine the PBLH, including new methods that have not been previously thoroughly evaluated. As such, this manuscript is acceptable to AMT pending minor revisions in which the following comments are adequately addressed.

We thank the Referee for the positive and constructive comments. We hope we have addressed all of the Referee's concerns and we think that our manuscript did benefit from the constructive comments made by all Referees.

Specific Comments

Title: Suggest changing name from 'Evaluating daytime planetary …' to 'Evaluating convective planetary'… given the focus is on convective PBLH estimates with the parcel method, which would be inappropriate for stable PBLH estimates.

We accepted the Referee's suggestion and changed the title accordingly.

Line 24 & 62: Should this be 'Collaborative Lower Atmospheric Mobile Profiling Systems'? That is the name given for it on the NSSL website and seems more apt.

We thank the Referee for catching the typo.

Line 177: While radiosondes are used as the truthing dataset here and are treated as the 'gold standard' (which is fine), there are still limitations in radiosondes for use of determining the PBLH. Most notably is that radiosondes provide a nearly instantaneous measurement and are only representative of the exact location it transited the BL / free troposphere interface. Thus, if the radiosonde transited this interface at a downdraft, the local PBLH estimate may be slightly

low biased compared to the area-averaged PBLH. Conversely, if an updraft is present the local PBLH would be slightly displaced upward compared to the area averaged PBLH. Over a large number of profiles, these effects should average out thus not leading to a bias. Still, it may lead to some of the significant scatter seen in the comparison plots later on.

We agree with the Referee's comment, therefore we decided to add a mention to the limitations in the use of radiosondes for determining the PBLH to the revised manuscript, essentially paraphrasing the reviewer's own very clear description of this limitation (Section 2.1, before the description of the various radiosonde methodologies).

Line 187 (and the entire manuscript): Highly recommend removing the radiosondes at 6 LT from the analysis. As the authors clearly state here, the parcel method is suited for convective conditions and the PBL is rarely convective at 6 LT, especially during the fall when sunrise is later.

According to all Referees' comments, we removed the radiosondes at 06:00 LT from the analysis. At the beginning of Section 2.1 we included the text *"Since sunrise for 19-24 August and 23-28 September is approximately at 05:20 and 06:00 LT, respectively, the 06:00 LT sounding was not included in the analysis as a convective PBL would not yet have been present in the remote sensing observations. Sunset for the 19-24 August and 23-28 September is at around 19:00 and 18:00 LT, respectively."* As a consequence of this change, all figures have been reproduced and all statistical results have been updated in the revised version of the manuscript.

Line 221: Is 'TROPoe' an acronym for something?

TROPoe is the acronym for Tropospheric Remotely Observed Profiling via Optimal Estimation. This has now been included in the body of the manuscript (Section 2.2.1).

Line 254: Is there evidence that supports that inclusion of RASS or model data in the TROPoe retrieval does not improve PBLH estimates? A short statement on the impact would suffice, or an appendix if additional analysis is warranted.

In addition to the one set of TROPoe-derived thermodynamic profiles constructed using only the passive instrument, we also derived other TROPoe thermodynamic profile sets (using the passive instrument plus RASS, and using the passive instrument plus the Rapid Refresh model, RAP). These are all archived to the NCAR/EOL data repository. We then performed the same analysis presented in the study but using the TROPoe retrievals that include the RASS and those that include the RAP, and we found no impact (or very minimal) in the PBLH estimates, probably because the range of heights covered by the RASS does not reach the top of the boundary layer most of the time, and the RAP is only assimilated well above the PBLH (> 4km agl). For this reason, we decided to mention this finding, but to not show the results in the manuscript. In the revised version of the manuscript we clarified that this was our finding *"RASS [...] we found its inclusion to not significantly impact the PBLH estimates (not shown). Similarly, while outputs from a numerical weather prediction model may outperform traditional retrieval methods for temperature and humidity profiling (Cimini et al., 2011), we found this inclusion to not*

*significantly impact the PBLH estimates (not shown). Therefore, the retrieved profiles including RASS and RAP input are not used in this study."*

Line 269: What range gate size was used for the Doppler lidar measurements? This detail can help the reader interpret the results given the tradeoff between reduced range/sensitivity (for a shorter gate) and reduced ability to resolved small turbulent eddies (for a longer gate).

The instrument specifications for the Doppler lidar measurements associated with the CLAMPS are of a range gate configurable in the 18-60 m range of values. Settings for the data collected by the vertical stare of the CLAMPS at Lakeland and Prentice, have a range gate of 45 and 30 m, respectively. This info has been added to the revised version of the manuscript in Section 2.3.1.

Line 319: It would be helpful to give a brief 1-2 sentence description of the Mues et al (2017) algorithm to summarize how it work. How is each 16-s PBLH estimate established independently from the backscatter profile?

As suggested by the Referee we expanded this section describing the differences between our approach and the Mues et al. (2017) approach, providing more details. Also, the Vaisala's BL-View software provides high-temporal resolution (16-s) estimates of the PBLH based on the assumption that within the mixed layer the aerosol concentration is vertically nearly constant and decreases above. This has been included in Sect. 2.3.2.

Figure 5: Perhaps I'm misunderstanding something, but the BL-View 1-h mean PBLH doesn't look like a mean of the individual 16-s PBLH estimates. This is particularly apparent between 7-8 LT at the Lakeland site, where most of the 16-s measurements show the BLH being around 400 m whereas the 1-h mean is at about 1 km. Please explain.

The BL-View software calculation of the hourly PBLH estimates does resemble more a median than a mean calculation, with the 200 m distance restriction avoiding unrealistic results in two-layer situations (Mues et al., 2017). Nevertheless, we agree with the Referee that the light-blue diamond estimates at hours 06:30 and 07:30 LT are not intuitively easy to understand. Unfortunately, since this is the provided hourly proprietary-software output we cannot track down its calculation. On the other hand, our QC-scaled PBLH estimates at the same time (orange dots) seem more reasonable, so that its behavior in cases like this made us comfortable in using it for our study.

Figure 6 discussion: Why does it look like there's a high bias in the QC-scaled PBLHs that scales with the PBLH itself? This is particularly apparent at the Lakeland site, wherein many of the QC-scaled PBLH measurements are above the 1-1 line when the PBLH is greater than 2.5 km.

We thank the Referee for catching this source of confusion. We accidentally forgot to include the 1-to-1 line in panel a) of Fig. 6 (Lakeland), therefore it seems indeed that there is a high bias in PBLH estimated using the QC-scaled approach for PBLH > 2.5 km. But in reality, most of the points do fall on the 1-to-1 line (black line now added to panel a), while the grey line represents the best-fit line, which has a slope < 1 due to the outliers in the right lower corner of the figure.

Additionally, the wording *"Black lines represent the 1-to-1 line, and grey lines represent the best-fit line"* has been included in the caption of Fig. 6 and the best-fit line equations have been included inside the panels (and elsewhere, see below).

Line 393: While I understand having an independent 'expert' dataset is useful for assessing the PBLH, the estimates are subjective. I myself would place the PBLHs differently in Fig.8, as the clouds between 10-15 LT appear to be cumulus clouds (could be verified with visible satellite imagery) meaning the PBLH would be higher than currently indicated and at least at the cloud base. One way to get around this subjectivity would be to have each coauthor provide independent estimates of the PBLH each hour that could be averaged together. Given these estimates are for only 2 IOPs (each a week long), this shouldn't be a lot of work for each coauthor.

We accepted the Referee's suggestion and consequently involved other co-authors (4 in total) in the independent PBLH 'expert' estimations. These independent estimations were then averaged and used in the revised version of the manuscript (unless the difference between them exceeded 300 m, in which case no PBLH estimate was provided). Text describing this change was added to the revised version of the manuscript. Fig. 8 has been updated with these new averaged estimates. Note that the PBLH estimate at hour 12:00 LT in Fig. 8 is now missing as the differences in estimates between the 'experts' exceeded 300 m. Statistical results have also been updated in the revised version of the manuscript (Figs. 12 and 13 and Table 4) using these new averaged estimates. Nevertheless, the overall conclusions still hold true.

Figure 9 (and elsewhere): It would be useful to provide the equation of the best-fit line for each subpanel. I assume the grey line shows this (it's unclear since it's not explained in the caption).

The wording *"Black lines represent the 1-to-1 line, and grey lines represent the best-fit line"* has been included in all figure captions, when necessary (Figs. 6, 9, 11, 12, and 13). Also, the best-fit line equations have been included inside the panels as suggested.

Line 437: What exactly is meant by 'time-matched'? It's unclear how these comparisons are different from those that are not time-matched.

We tried to clarify the meaning of "time-matched" in the revised version of the manuscript (in the same place pointed by the Referee) by including the wording *"To adequately compare the statistical results obtained by the different instruments, the analysis was also performed only on the concurrent (i.e., time-matched) PBLH estimates, only when values are present from all instruments at a given site."*
Conversely, the non-time-matched statistical results include all estimations from one instrument, regardless of the availability or not of the other instruments' estimations.

Lines 527-553: This is a very long paragraph. Suggest breaking up into at least 2, perhaps 3 paragraphs for readability.

We divided this paragraph into smaller ones as suggested.

Line 597: For this profile at 13:00, the PBLH in the radiosonde profile is relatively ambiguous compared to other times (on this day and for the summer case). This is due to the fact that there is a slightly stable layer between 1.5 and 2.5 km. Thus, small changes in measurements of the surface temperature (due to instantaneously measuring warm or cold anomalies) as well as use of the 0.5 K addition in the PBLH can make significant differences in the actual parcel-based PBLH estimate. This should be discussed.

We completely agree with the Referee on this point. As a matter of fact, from 13:00 LT on, is the time when the largest differences among all measurements are observed and it is quite difficult to say which one of the measurements is providing the correct PBLH estimation. We think this is a good case to use, since it shows that the PBLH is really difficult to estimate on some days. We had originally included something about it in the manuscript, but we acknowledge that it was not enough. We have expanded the text in the revised version of the manuscript including *"At 13:00 LT, the PBL is well mixed up to around 1.5 km agl, but another slightly stable layer is visible between 1.5 and 2.5 km agl (Fig. 16d). The radiosonde places the PBLH somewhere in the cloud layer near 2.1 km agl, but it has to be recognized that in a case like this, small changes in the surface temperature measurements (due to instantaneous warm or cold anomalies) can result in significant differences in the parcel-based PBLH estimate, therefore making the results very uncertain."*

Technical Corrections:

Lines 22-23: These instruments should not be capitalized.

Agreed. Changed as suggested.

Line 75: i.e. should be followed by a comma (i.e.,)

Agreed. Changed as suggested.

Line 81: SNR should be defined as an acronym here where it first appears, not at line 90.

Agreed. Changed as suggested.

Line 202: Should the -1 should be superscript?

Agreed. Changed as suggested.

Line 499: Should CLAMPS be plural here (CLAMPSs)?

Agreed. Changed as suggested.

---

## Author Comment (AC2)

**Comment on amt-2021-363**

**Anonymous Referee #3**

Referee comment on "Evaluating daytime planetary boundary-layer height estimations resolved by both active and passive remote sensing instruments during the CHEESEHEAD19 field campaign" by James B. Duncan Jr. et al., Atmos. Meas. Tech. Discuss., https://doi.org/10.5194/amt-2021-363-RC2, 2022

The current manuscript about the Planetary Boundary Layer Height (PBLH) analyzes and compares retrievals from different methods and instruments during the CHEESEHEAD19 field campaign. This subject is within the agenda of AMT and is of high interest for the scientific community, since it is not common to have this number of instruments in a close range. The work focuses on the differences between methods and instruments and validates retrievals using collocated radiosondes for reference. Case studies of days with different cloud conditions offer a deeper insight of the inconsistencies between the retrievals. The manuscript is well written and all major issues of the methods and the results are discussed, hence I suggest to be accepted for publication after minor revisions.

We thank the Referee for the positive and constructive comments. We hope we have addressed all of the Referee's concerns and we think that our manuscript did benefit from the constructive comments made by all Referees.

**Specific comments:**

Introduction: I think some literature should be added, considering comparisons of retrievals from the instruments used in this study. Also, some discussion is expected about the different definitions of PBL and the known differences among the retrievals based on the variable in study.

**As suggested, some literature and discussion has been added to the Introduction.**

Section 2.1 Some discussion about the problems/ errors/uncertainties of the radiosondes retrievals should be added.

According to this Referee's and to Referee #1's comments, additional discussion on the limitations in the use of radiosondes for determining the PBLH has been added to the revised manuscript (Section 2.1, before the description of the various radiosonde methodologies): "We note that limitations in using radiosondes for determining the PBLH include that they provide nearly instantaneous measurements and are only representative of the exact location transited at the interface between the boundary layer and the free troposphere above. Moreover, if the radiosonde transited a downdraft, the local PBLH estimate may be biased slightly low compared to the area-averaged PBLH. Conversely, if an updraft is present, the local PBLH estimate would be displaced slightly upward compared to the area-averaged PBLH. However, given the large number of profiles used in this study, the impacts of updrafts and downdrafts on PBLH estimate

**should average out and not lead to a bias; albeit, this may have contributed to some of the scatter in the comparison plots presented later on."**

Figure 2. The errorbars and the outliers should be described at the caption. I am in doubt that this representation of the variation of each method is the most adequate, because ranges of more than 2km for PBLH can include all possible values. Probably a visualization of synchronous values would be more appropriate. Also, somewhere earlier in the manuscript, the sunrise/sunset LT for the 7 day IOP, in order to understand the low values at 6.00LT. If 6.00 LT is before sunrise or even shortly after, the parcel method is not applicable, since it is referring to convective conditions.

Description of the error bars used in the figure has been added to the caption of Fig. 2: "*The boxes show the interquartile range with the median indicated by the horizontal line and the whiskers extend to points that lie within 1.5 times the interquartile range of the lower and upper quartiles*".

Also, according to all Referees' comments, we removed the radiosondes at 06:00 LT from the analysis, and we included sunrise and sunset times for the 2 IOPs in the manuscript as suggested by the Referee. At the beginning of Section 2.1 we included the text "Since sunrise for 19-24 August and 23-28 September is approximately at 05:20 and 06:00 LT, respectively, the 06:00 LT sounding was not included in the analysis as a convective PBL would not yet have been present in the remote sensing observations. Sunset for the 19-24 August and 23-28 September is at around 19:00 and 18:00 LT, respectively." As a consequence of this change, all figures have been reproduced and all statistical results have been updated in the revised version of the manuscript.

Since 6:00 LT was the time when larger differences in the number of available PBHL estimations between the different radiosonde methods were observed in Fig. 2, and since for the other radiosonde launch times the available number of PBLH estimations is pretty similar among the methods, we believe that our representation is now adequate.

Figure 4. The peak around 2.00LT should be discussed in the corresponding paragraph. It appears a variation in wind conditions during this time, that leads the algorithm to recognize a stratification at higher height.

Since the peak mentioned by the Referee happens at nighttime, in stable conditions, shearinduced turbulence could have caused increased values of vertical velocity variance, which is mainly used to determine the PBLH estimate during nighttime. As a matter of fact, the vertical velocity plot indicates stronger downdrafts at around this time which might be related to a vertical mixing process.

L323. The description of the method of selecting value based on the score is not described clearly. How the criterion of  $\pm 200$ m came up?

As suggested by the Referee we expanded this section describing the differences between our approach and the Mues et al. (2017) approach, providing more details. Also, the 200 m range restriction was introduced by Mues at al. (2017) to avoid unrealistic results in two-layer situations, as explained in their paper.

L330. The BL software provides the higher value of 4km in many cases of 16s retrievals, is there any physical explanation, considering the atmospheric conditions, for this result?

We have to clarify that not all hours when the cyan diamonds are missing correspond to PBLHs estimated from the BL-View having a higher value of 4 km agl. We made this now clearer in the revised text: *"The BL-View software has an upper limit for PBLH values of 4 km agl.* Additionally, due to the proprietary QC methods imposed within the BL-View software, there are some hours when no PBLH value is provided." Nevertheless, since the BL-View software is proprietary, we are not completely aware of how it operates, therefore we are not able to explain why it provides higher values of the QC-scaled approach for some hours (i.e., 14:30 LT at both sites). However, when all the days were visually inspected we noticed that when these (rare) differences occurred, the QC-scaled approach appeared to provide more reasonable PBLH estimates than the hourly BL-View output, so we were comfortable in employing the QC-scaled approach (which we had control on) in our study.

L350-355 The different response of the comparison between BL and QC for the two sites should be discussed. Is there some local or systematic effect that could explain the worst statistics for Lakeland?

Unfortunately, we are unable to pinpoint the reason for the different statistical comparisons between the BL-View and QC-scaled approach at the Lakeland and Prentice sites due to either local or systematic effects. This consideration was added to the text of the revised manuscript.

L390 I think the idea of an independent dataset selected manually by visual inspection of the recordings can be a valid reference for evaluating the retrievals. Why don't include more data from other instruments for creating this reference databases, specially in cases of sharp gradients?

We agree with the Referee that a synergetic use of the instruments presented in this study would be the optimal way to achieve accurate PBLHs, since the strengths of the various platforms would be combined together. We nevertheless believe that this would be a further step in the study and we would rather postpone it to a future analysis, as it would undoubtedly become rather complicated and require a lengthy description.

L420-425. I cannot see radiation information been used, only the cloud information. Please restate or explain. Also, the cloud fraction is the cause of different development of convective Boundary Layer, but the result is not immediate, since some time is needed to propagate the effect to the layer. Hence I suggest to investigate the possibility of correlating the with cloud fractions in a wider time window. More specifically a window including the previous time steps will correlate better due the delay response.

Regarding the cloud fraction information, we have clarified its use in the revised version of the manuscript rewording the text to *"Two RadSys stations were deployed at Lakeland and Prentice (data available from Riihimaki, et al., 2020a, b) providing complete surface irradiance measurements, which then enable RadFlux analysis (Long and Ackerman 2000; Long et al.*

2006) that produces derived cloud variables such as cloud fraction. This cloud fraction information was used to expand the PBLH evaluation under different cloud coverage conditions."

We agree with the Referee that the impact of the clouds on the convective boundary layer development can indeed be delayed, but we also think that it is difficult to establish what optimal time window to use that would be overall adequate for different cloud heights, different cloud types and cloud fractions, and for different cloud advections. Therefore, we decided to keep the window used to determine the cloud fraction as the same temporal window used to determine the PBLH estimates, but we included some text in the revised version of the manuscript mentioning the fact that this will indeed neglect taking into account the delayed impact of the clouds on the development of the convective boundary layer: *"The hour window used to determine the cloud fraction is the same temporal window* ( $\pm 30$  min of the hour) used to determine the PBLH estimates. This neglects any possible delays in the impact of clouds on the development of the convective boundary layer, which is however difficult to quantify."

Figure 9, The caption should explain what are the black and grey lines.

The wording "*Black lines represent the 1-to-1 line, and grey lines represent best-fit line*" has been included in all figure captions, when necessary (Figs. 6, 9, 11, 12, and 13). Also, the best-fit line equations have been included inside the panels.

---

## Author Comment (AC3)

**Comment on amt-2021-363**

Anonymous Referee #2

Referee comment on "Evaluating daytime planetary boundary-layer height estimations resolved by both active and passive remote sensing instruments during the CHEESEHEAD19 field campaign" by James B. Duncan Jr. et al., Atmos. Meas. Tech. Discuss., https://doi.org/10.5194/amt-2021-363-RC3, 2022

The manuscript "Evaluating daytime planetary boundary-layer height estimations resolved by both active and passive remote sensing instruments during the CHEESEHEAD19 field campaign" by James B. Duncan Jr. et al. efficiently analyzes the daytime evolution of the planetary boundary-layer, by using multiple active and passive remote sensing instruments, as well as radiosonde observations. It is well organized, meets scientific quality and provides valuable information within the scope of the Atmospheric Measurement Techniques community. Moreover, the figures gather important information that is featured clearly.

We thank the Referee for the positive and constructive comments. We hope we have addressed all of the Referee's concerns and we think that our manuscript did benefit from the constructive comments made by all Referees.

Specific comments:

Since parcel method is better suited in convective boundary layer conditions, please comment the fact that it is employed to derive PBLH at 6 local time, when convection is not accomplished.

According to all Referees' comments, we removed the radiosondes at 06:00 LT from the analysis. At the beginning of Section 2.1 we included the text *"Since sunrise for 19-24 August and 23-28 September is approximately at 05:20 and 06:00 LT, respectively, the 06:00 LT sounding was not included in the analysis as a convective PBL would not yet have been present in the remote sensing observations. Sunset for the 19-24 August and 23-28 September is at around 19:00 and 18:00 LT, respectively."* As a consequence of this change, all figures have been reproduced and all statistical results have been updated in the revised version of the manuscript.

Figure 4: Please add a label for "Prentice" and "28 September 2019", like in figures 3,5,7,8. It is really helpful to see this information pointed out, considering the amount of data and case-studies.

As suggested, the site and date labels have been added to Fig. 4.

Figure 2: A more detailed caption, including the error bar and outlier information would be supportive for the reader.

Description of the error bars used in the figure has been added to the caption of Fig. 2: *"The boxes show the interquartile range with the median indicated by the horizontal line and the*

*whiskers extend to points that lie within 1.5 times the interquartile range of the lower and upper quartiles.”*